# Genetic and environmental perturbations lead to regulatory decoherence

**Amanda Lea[1,2†], Meena Subramaniam[3†], Arthur Ko[4], Terho Lehtimäki[5,6], Emma Raitoharju[6], Mika Kähönen[6,7], Ilkka Seppälä[6], Nina Mononen[6], Olli T Raitakari[8,9], Mika Ala-Korpela[10,11,12,13,14,15], Päivi Pajukanta[16], Noah Zaitlen[3‡], Julien F Ayroles[1,2‡*]**

[1]Department of Ecology and Evolution, Princeton University, Princeton, United States; [2]Lewis-Sigler Institute for Integrative Genomics, Princeton University, Princeton, United States; [3]Department of Medicine, Lung Biology Center, University of California, San Francisco, San Francisco, United States; [4]Department of Medicine, David Geffen School of Medicine at UCLA, University of California, Los Angeles, Los Angeles, United States; [5]Department of Clinical Chemistry, Fimlab Laboratories, Faculty of Medicine and Health Technology, Tampere University, Tampere, Finland; [6]Finnish Cardiovascular Research Center, Faculty of Medicine and Health Technology, Tampere University, Tampere, Finland; [7]Department of Clinical Physiology, Tampere University, Tampere University Hospital, Tampere, Finland; [8]Research Centre of Applied and Preventive Cardiovascular Medicine, University of Turku, Turku, Finland; [9]Department of Clinical Physiology and Nuclear Medicine, Turku University Hospital, Turku, Finland; [10]Systems Epidemiology, Baker Heart and Diabetes Institute, Melbourne, Australia; [11]Computational Medicine, Faculty of Medicine, Biocenter Oulu, University of Oulu, Oulu, Finland; [12]NMR Metabolomics Laboratory, School of Pharmacy, University of Eastern Finland, Kuopio, Finland; [13]Population Health Science, Bristol Medical School, University of Bristol, Bristol, United Kingdom; [14]Medical Research Council Integrative Epidemiology Unit, University of Bristol, Bristol, United Kingdom; [15]Department of Epidemiology and Preventive Medicine, School of Public Health and Preventive Medicine, Faculty of Medicine, Nursing and Health Sciences, The Alfred Hospital, Monash University, Melbourne, Australia; [16]Department of Human Genetics, David Geffen School of Medicine at UCLA, University of California, Los Angeles, Los Angeles, United States

**\*For correspondence:**
jayroles@princeton.edu

†These authors contributed equally to this work
‡These authors also contributed equally to this work

**Competing interests:** The authors declare that no competing interests exist.

**Abstract** Correlation among traits is a fundamental feature of biological systems that remains difficult to study. To address this problem, we developed a flexible approach that allows us to identify factors associated with inter-individual variation in correlation. We use data from three human cohorts to study the effects of genetic and environmental variation on correlations among mRNA transcripts and among NMR metabolites. We first show that environmental exposures (infection and disease) lead to a systematic loss of correlation, which we define as 'decoherence'. Using longitudinal data, we show that decoherent metabolites are better predictors of whether someone will develop metabolic syndrome than metabolites commonly used as biomarkers of this disease. Finally, we demonstrate that correlation itself is under genetic control by mapping hundreds of 'correlation quantitative trait loci (QTLs)'. Together, this work furthers our understanding of how and why coordinated biological processes break down, and points to a potential role for decoherence in disease.

**Editorial note:** This article has been through an editorial process in which the authors decide how to respond to the issues raised during peer review. The Reviewing Editor's assessment is that all the issues have been addressed (see decision letter).

DOI: https://doi.org/10.7554/eLife.40538.001

## Introduction

A major goal in evolutionary and medical genetics is to identify the coordinated regulatory processes that differ between individuals as a function of their disease state, environment, or genetic background. One powerful approach for doing so is to identify the effect of environmental or genetic perturbations on the degree to which genes are correlated at the mRNA level – a phenomenon known as 'co-expression'. From a medical perspective, identifying factors associated with changes in co-expression between healthy and sick individuals should point toward key regulatory changes driving phenotypic differences between groups (*Mentzen et al., 2009*; *Hudson et al., 2009*; *Kostka and Spang, 2004*; *Choi et al., 2005*; *de la Fuente, 2010*). From an evolutionary perspective, work on decanalization (*Gibson, 2009a*; *Gibson, 2009b*; *Careau et al., 2014*) indicates that gene regulatory networks evolve over many generations of stabilizing selection, and new mutations or novel environments may disrupt these fine-tuned connections. Decanalization has been hypothesized to explain the recent rise of non-communicable diseases in humans (*Gibson, 2009b*; *Hu et al., 2016*), with the idea being that recent major shifts in diet and lifestyle have led to dysregulation of regulatory programs that evolved under past environmental conditions. However, testing this hypothesis remains challenging in practice.

Our ability to move forward in understanding how environmental and genetic variation affect molecular correlations – in both humans and other organisms where this topic has been extensively discussed (*Wang et al., 2013*; *Sun et al., 2008*; *Southworth et al., 2009*; *Fukushima, 2013*; *Huang et al., 2015*) – is limited by the available methodology (reviewed in (*de la Fuente, 2010*; *van Dam et al., 2017*; *Singh et al., 2018*)). In particular, most studies of differential co-expression to date have relied on two types of approaches: (i) building co-expression networks separately within each group of interest (e.g., diseased versus healthy) and contrasting their properties (*Kostka and Spang, 2004*; *Choi et al., 2005*; *Steiger, 1980*; *Watson, 2006*); or (ii) asking whether predefined sets of genes are differentially co-expressed between groups (*Fukushima, 2013*; *Jen et al., 2006*). Generally, these approaches are designed to compare population-level statistics (e.g., correlation coefficients) between two sample groups, and thus cannot test the effect of continuous predictor variables on variation in correlation. Further, these approaches are generally designed for datasets in which no individual-level covariates (e.g., age, sex, or batch) may bias co-expression estimates, which could lead to false conclusions if not accounted for (*Parsana, 2017*). This limited flexibility has made it difficult to identify individual-specific factors that predict co-expression in humans or other organism that are not amenable to controlled manipulations.

To address this gap, we present a new approach for interrogating sources of variance in correlation. Our approach builds on previous work (*Steiger, 1980*), and is based on the fact that the Pearson correlation coefficient is equal to the average element-wise product of two traits measured across individuals, after each trait is mean centered and scaled; this value reflects the relationship between two traits within a population sample. By extension, to obtain a measure of the degree of correlation between two traits *for each individual* in a sample, we can simply keep the vector of products (i.e., we do not perform averaging across individuals). This vector of products can be used as the outcome variable in a linear model or linear mixed model, and consequently our approach can accommodate covariates and continuous predictor variables. We call our approach '*Correlation by Individual Level Product*' (CILP), because for each individual the product between two traits is estimated and modeled as a continuous outcome variable (*Figure 1*).

Leveraging this approach, we explore the effects of environmental and genetic variation on the degree to which two molecular traits are correlated. First, we test the hypothesis that stressful physiological conditions (here, bacterial infection or metabolic syndrome) lead to a loss of correlation among molecular traits that are correlated under normal conditions. We refer to this loss of correlation in an infected or diseased state, relative to a baseline or healthy state, as 'decoherence'. We test for decoherence using gene expression data derived from monocytes at baseline or following

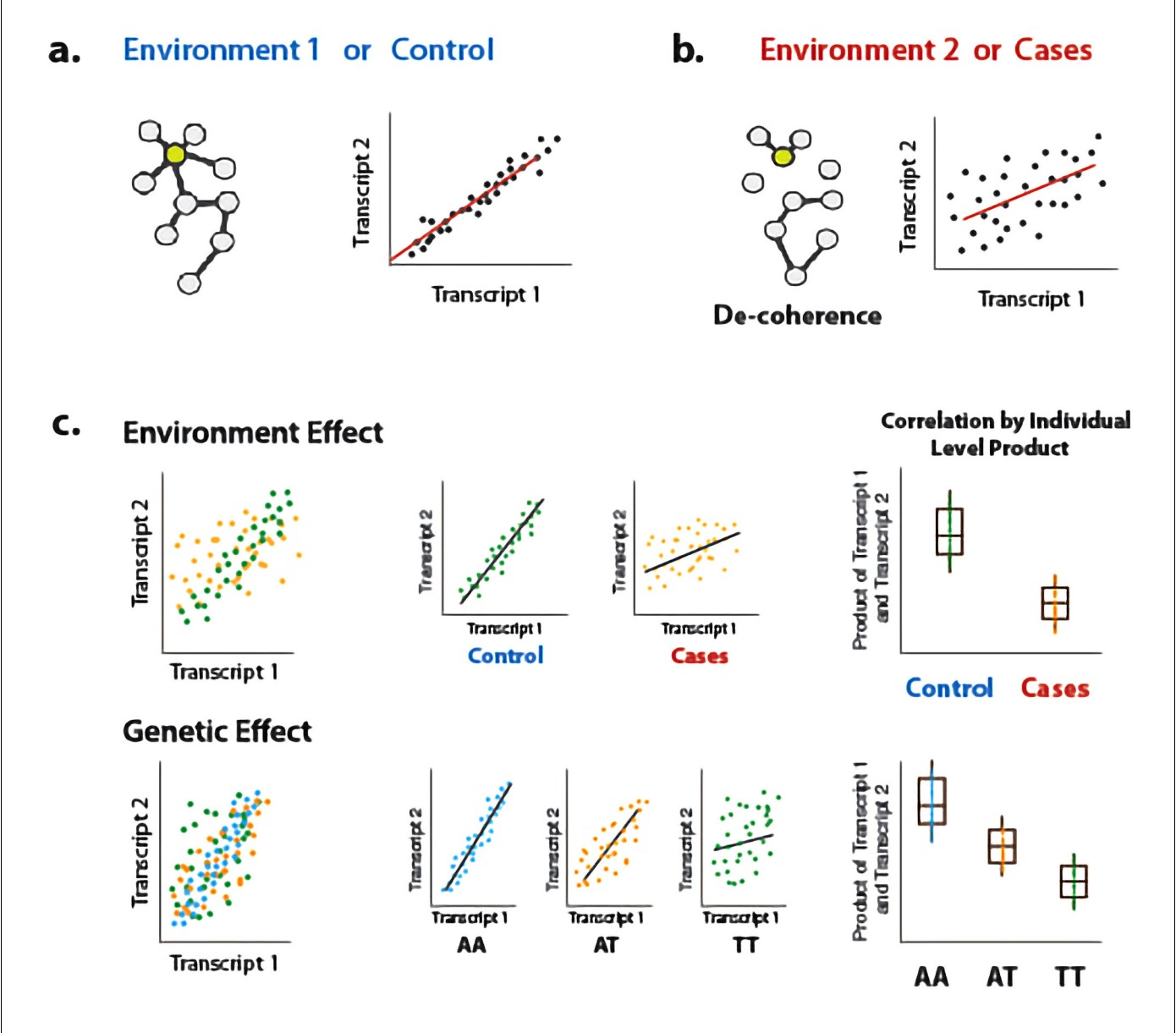

**Figure 1.** Illustration of decoherence and 'Correlation by Individual Level Product' (CILP). (a) Co-expression networks in two different environments. Environment one represent normal (or control/baseline) conditions where the expression levels of gene 1 and gene 2 are highly correlated and co-regulated. (b) Environment 2 represents a stressful or unhealthy condition leading to lower correlation between the expression levels of gene 1 and gene 2. At the network level, this translates into a lower network degree (i.e., lower average correlation across genes or fewer connected nodes). We call this change in the correlation structure 'decoherence'. This is what we could expect if stressful conditions lower transcriptional robustness and lead to dysregulation of gene expression. (c) Differences in correlation between the expression levels of gene 1 and gene 2 in cases versus controls, or between genotypes, which translates into an average difference in product between these groups.

DOI: https://doi.org/10.7554/eLife.40538.002

The following figure supplements are available for figure 1:

**Figure supplement 1.** Simulations reveal power to detect correlation QTLs across a range of effect sizes.
DOI: https://doi.org/10.7554/eLife.40538.003

**Figure supplement 2.** Simulated batch effects on transcriptional correlation structure do not produce false positive correlation QTLs.
DOI: https://doi.org/10.7554/eLife.40538.004

stimulation with lipopolysaccharide (*Fairfax et al., 2014*), a component of bacterial cells walls and a potent stimulant of the innate immune response (*Fairfax et al., 2014*). We also test for decoherence using blood-derived NMR metabolite data from the Young Finns Study (YFS) (*Raitakari et al., 2008*). Strikingly, we find strong evidence in both datasets. Finally, we use CILP to map 'correlation QTL', defined as SNPs that affect the magnitude of the correlation between two mRNA transcripts. Using genotype and whole blood-derived gene expression data from the Netherlands Study of Depression and Anxiety (*Penninx et al., 2008*), we identity and replicate hundreds of correlation QTLs. Together, our new approach allows us to identify genetic variants and environmental factors that disrupt molecular co-regulation. Further, the flexible, robust approach we propose opens the door to future investigations of the causes and consequences of trait covariation in many contexts.

## Results

### Using 'Correlation by Individual Level Product' (CILP) to test for sources of variance in correlation

Let $y_1$, $y_2$ be two outcomes measured across individuals in a population sample, with means $\bar{y}_1$, $\bar{y}_2$ and variances $\sigma_1^2, \sigma_2^2$, respectively. We wish to associate the Pearson correlation coefficient between these two variables, $E[(y_1 - \bar{y}_1)(y_2 - \bar{y}_2)]/\sqrt{\sigma_1^2 \sigma_2^2}$, with some predictor variable $x$. We propose the following statistical test. First, we calculate the demeaned, element-wise product of the outcomes: $[(y_1 - \bar{y}_1)(y_2 - \bar{y}_2)]$, and then normalize by the square root of the product of variances: $\left[(y_1 - \bar{y}_1)(y_2 - \bar{y}_2)/\sqrt{\sigma_1^2 \sigma_2^2}\right]$. The resulting vector of values represents the estimate of the correlation within each individual in the sample, which can subsequently be modeled using approaches appropriate for continuous outcome variables. In practice, we use linear models or linear mixed effects models to test for associations between a given set of products and a fixed effect predictor variable, while controlling for covariates. We note that for our correlation QTL analyses, described later in the text, we use a modified version of CILP for computational efficiency and scalability. This modified version is described in the section '*Comparison of approaches for quantifying correlation*' (see also *Supplementary file 1*).

### Simulations reveal power to detect sources of variance in correlation across many scenarios

To confirm that our approach does not result in biased p-value distributions, and to understand the power of CILP across a range of effect sizes and sample sizes, we first performed extensive simulations. We focused our simulations on the identification of 'correlation QTL', defined as SNPs that affect the magnitude of the correlation between two mRNA transcripts; however, we note that the results are generalizable to other predictor variables. For each simulation, we generated 10,000 gene pairs that differed in covariance as a function of genotypic class. We did so using the multivariate normal distribution to simulate pairs of continuous distributions (but see *Figure 1—figure supplement 1* for results where count data were simulated from a negative binomial distribution). Following simulation of gene expression data, we used CILP and linear models to detect differences in correlation as a function of genotype.

With an effect size of 0.3 and n = 1000, we detected 98.19% of simulated true positive correlation QTLs at a threshold of p=0.05, and 50.91% of true positives after correcting for multiple hypothesis testing (using Bonferroni correction). Under the null, where the effect size was set to zero, we detect 5.19% of correlation QTLs at a threshold of p=0.05 and no correlation QTL after correcting for multiple testing (*Table 1*, *Figure 1—figure supplement 1*). To ensure that changes in the mean, variance, or covariance of gene expression levels do not increase our false positive rate, we also assessed power to detect correlation QTLs when: (i) the focal SNP affected mean gene expression levels (e.g., through an eQTL or proportion QTL); (ii) the focal SNP affected the variance of gene expression levels (e.g., through a varQTL); (iii) the mean or variance of gene expression levels were affected by a variable unrelated to genotype (e.g., through batch, cell type, or environmental effects); and (iv) a variable unrelated to genotype affected the correlation structure of the entire transcriptome. None of these scenarios increased the false positive rate, though the presence of strong varQTLs did decrease power to detect correlation QTLs (*Table 1*, *Figure 1—figure supplement 2*). We also

**Table 1.** Power to detect simulated correlation QTL across a wide range of scenarios (n = 1000 for all simulations).

**Simulation parameters**

| | | | | | | | | |
|---|---|---|---|---|---|---|---|---|
| (1) Correlation QTL effect[*] | x | x | x | x | x | x | x | x |
| (2) Genotype predicts the mean of gene one expression levels[†] (e.g., through an eQTL or proportion QTL) | | x | x | | x | | | |
| (3) Genotype predicts the mean of gene two expression levels[†] | | | x | | | | | |
| (4) Genotype predicts the variance of gene one expression levels[‡] (e.g., through a varQTL) | | | | x | x | | | |
| (5) A variable that is random with respect to genotype predicts the mean of gene one expression levels[†] (e.g., through technical, environmental, or cell type heterogeneity effects) | | | | | | x | | |
| (6) A variable that is random with respect to genotype predicts the mean of gene two expression levels[†] | | | | | | x | | |
| (7) A variable that is random with respect to genotype predicts the variance of gene one expression levels[‡] (e.g., through technical, environmental, or cell type heterogeneity effects) | | | | | | | x | x |
| (8) A variable that is random with respect to genotype predicts the variance of gene two expression levels[‡] | | | | | | | | x |

**Results**

| | | | | | | | | |
|---|---|---|---|---|---|---|---|---|
| Power (proportion of true positives detected, Bonferroni-corrected p<0.05) | 0.5091 | 0.4934 | 0.5055 | 0.0716 | 0.0679 | 0.4998 | 0.0884 | 0.0533 |
| False positive rate (proportion of true negatives detected, nominal p<0.05) | 0.0519 | 0.0529 | 0.0494 | 0.0532 | 0.0507 | 0.0527 | 0.0529 | 0.0522 |

Simulated effect sizes were as follows: correlation QTL[*] = 0.3, mean effects[†] = 1, variance effects[‡] = 10. When Bonferroni-corrected p-values were used, the false positive rate was 0 across all simulation scenarios. Abbreviations: eQTL = expression QTL and vQTL = variance QTL.

DOI: https://doi.org/10.7554/eLife.40538.005

found that including the expression levels of both genes as covariates in our models did not improve our power to detect correlation QTLs (*Figure 1—figure supplement 1*).

## Immune challenges results in decreased co-regulation of gene expression

Next, we used gene expression data collected from human monocytes (*Fairfax et al., 2014*), to ask whether patterns of co-expression differed between cells assayed at baseline versus following treatment with LPS for 2 or 24 hr (n = 214 samples were assayed across the three conditions). To limit our search space, we focused on 1460 genes that were differentially expressed at the 2 hr time point (FDR < 5%), and tested for differences in correlation between unexposed and LPS-exposed cells across 1,063,611 possible transcript pairs (equivalent to 1460 chose 2). We found 958 gene pairs with a significant (FDR < 5%) change in correlation between the uninfected/baseline state versus 2 hr post LPS stimulation, 461 of which replicated with similar effect sizes at the 24 hr time point (correlation between effect sizes at the two time points: $R^2$ = 0.12, $p<10^{-16}$, binomial test for concordance of effect size direction: $p<10^{-16}$; *Figure 2*).

We observed no relationship between the magnitude of the difference in mean gene expression levels between conditions and the magnitude of the difference in correlation between conditions, suggesting our results are not driven by statistical artifacts associated with large mean changes in transcription following LPS stimulation (*Figure 2—figure supplement 1*). Related to this point, we found minimal overlap (<5% for all comparisons) between genes with significant infection-associated variation in correlation and (i) genes that were differentially expressed after 2 or 24 hr of LPS treatment (FDR < 5%) or (ii) genes with eQTL that were specific to the baseline, 2 hr LPS treatment, or 24 hr LPS treatment samples (as reported in *Fairfax et al. (2014)*).

In total, we identified gene pairs that are more highly correlated in cells assayed post LPS stimulation versus at baseline, as well as gene pairs that lose correlation following a simulated bacterial infection. Overall, however, we found much greater support for the latter pattern: 61% (at the 2 hr time point) and 73% (at the 24 hr time point) of significant transcript pairs were more strongly correlated across individuals in uninfected versus LPS challenged cells. Importantly, this represents a statistically significant bias toward a loss of correlation (i.e. decoherence) following two hours of simulated bacterial infection ($p=1.05\times10^{-7}$, $\log_2$ odds = 0.503, Fisher's exact test), with an even

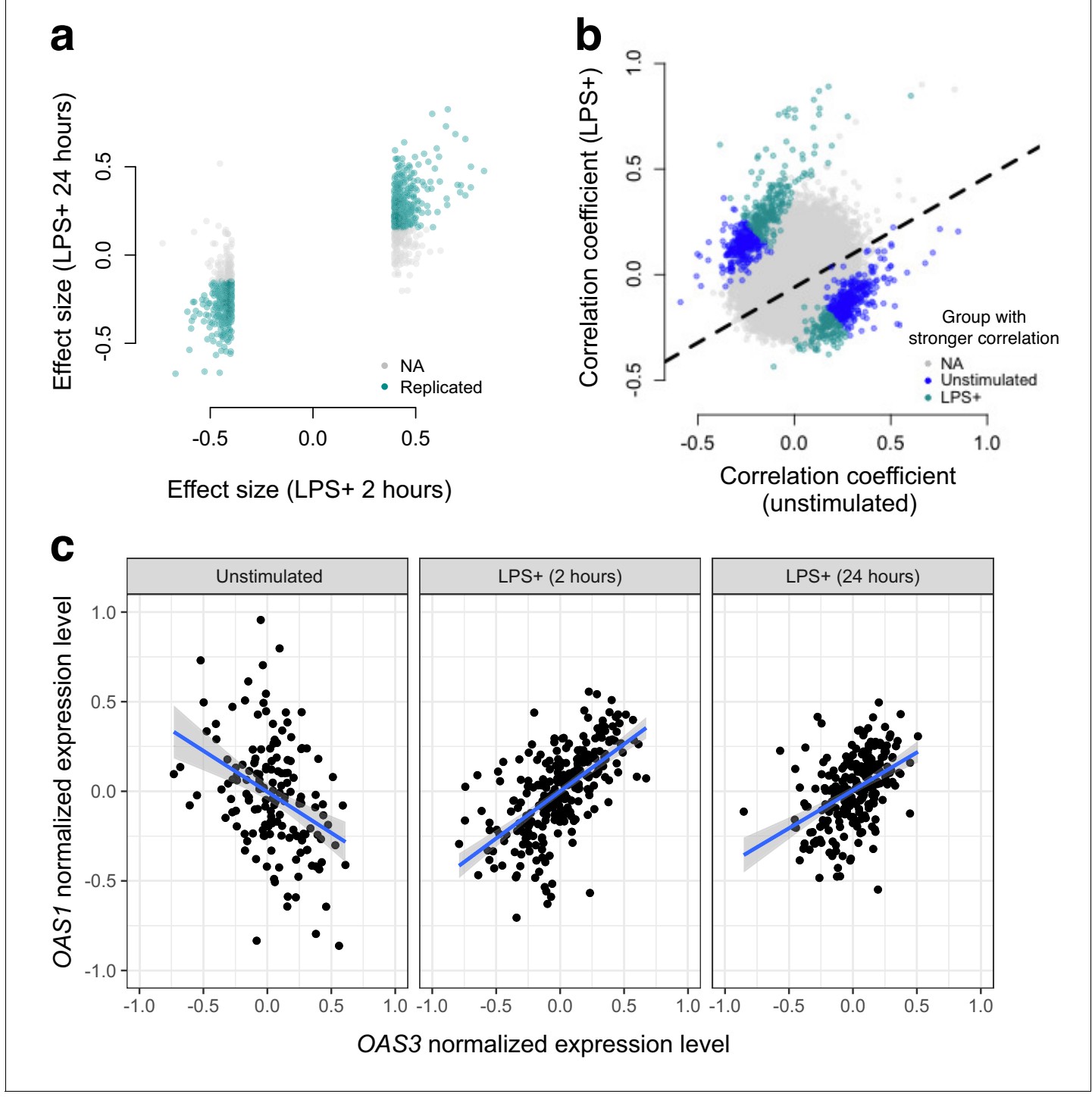

**Figure 2.** Bacterial infection (i.e., treatment with LPS) leads to decoherence in primary monocyte gene expression. (a) Correlation changes have similar effect sizes after 2 hr or 24 hr of LPS stimulation. (b) Comparison of the Spearman correlation coefficient estimated in unstimulated cells (x-axis) versus those treated with LPS (y-axis). Gene pairs for which we detect significantly stronger correlations in the unstimulated (blue dots) or LPS treated (green dots) condition are highlighted. (c) An example of changes in correlation as a function of condition. Gene expression levels for OAS1 and OAS3 are negatively correlated at baseline, but become positively correlated after 2 hr or 24 hr treatment with LPS.

DOI: https://doi.org/10.7554/eLife.40538.006

The following figure supplement is available for figure 2:

**Figure supplement 1.** Genes that are strongly differentially expressed between conditions (unstimulated versus LPS 2 hr) are not more likely to change their correlation structure.

*Figure 2 continued on next page*

*Figure 2 continued*

DOI: https://doi.org/10.7554/eLife.40538.007

stronger bias toward decoherence after 24 hr of immune challenge ($p<10^{-16}$, $\log_2$ odds = 0.941, Fisher's exact test).

## Metabolic syndrome disrupts metabolite co-regulation

### Identification of disease-associated variation in metabolite correlations

We next applied our correlation test to whole blood-derived NMR metabolite data collected from a population-based, longitudinal study of young Finnish individuals (the cardiovascular risk in young Finns study, abbreviated 'YFS') (*Raitakari et al., 2008*; *Nuotio et al., 2014*). Our dataset included 159 metabolite measures collected across three time points that spanned a 10 year period (2001: n = 1564, 2007: n = 1498, 2011: n = 1501). We note, however, that our analyses did not focus on changes in correlation across time, but rather, changes in correlation between individuals that were healthy versus those that met the criteria for metabolic syndrome (*Grundy et al., 2005*).

To test the hypothesis that disease disrupts homeostasis and leads to decoherence, we asked whether specific pairs of metabolites were correlated in healthy individuals, but no longer correlated in individuals with metabolic syndrome. Across 11491 unique metabolite pairs that met our criteria for analysis (see Materials and methods), healthy individuals and those with metabolic syndrome showed broadly similar correlation structures (*Figure 3*). However, using CILP, we identified strong effects of health status on the degree of correlation between a subset of metabolite pairs. Specifically, we identified 1528 metabolite pairs that were more correlated in healthy individuals relative to those with metabolic syndrome, and 619 metabolite pairs that showed the opposite pattern (FDR < 5%; linear mixed model controlling for age, sex, year, and individual identity). This represents a 2.20x enrichment ($p<10^{-16}$, Fisher's exact test) of metabolite pairs that lose correlation following the onset of disease. Importantly, the set of 1528 metabolite pairs that lose correlation includes metabolites that are both more highly expressed (n = 93 metabolites, FDR < 5%) and more lowly expressed (n = 32) in individuals with metabolic syndrome, as well as metabolites that do not significantly differ between sample groups (n = 4).

### Metabolic syndrome affects co-regulation of particular metabolite classes

We found overall support for the hypothesis that metabolic syndrome disrupts the correlation structure that exists in healthy individuals. To identify the specific pathways and processes targeted by metabolic syndrome, we assigned each metabolite in our dataset to one of 11 functional classes (as in *Nath et al. (2017)*; *Supplementary file 2*), and asked whether each class was enriched for metabolites that exhibited decreases in correlation. Here, we found the strongest enrichment for apolipoproteins (hypergeometric test, odds ratio = 1.94, $p=2.34\times10^{-13}$), measures of total cholesterol (odds ratio = 1.39, $p=1.58\times10^{-15}$), and small molecules involved in energy metabolism (odds ratio = 1.43, $p=2.75\times10^{-13}$). Among metabolite pairs that showed the opposite pattern (i.e., were more strongly correlated in individuals with metabolic syndrome), we found an enrichment of fatty acids (odds ratio = 1.30, $p=7.10\times10^{-6}$), as well as HDL (odds ratio = 1.18, $p=1.53\times10^{-5}$) and LDL lipoproteins (odds ratio = 1.28, $p=3.58\times10^{-6}$) (*Figure 3*; *Supplementary file 3*). Together, these results suggest that metabolite co-regulation is strongly perturbed by disease, and further, that particular classes of metabolites are more sensitive and prone to decoherence than others.

Finally, because our dataset included metabolite data collected from the same individuals across multiple time points, we asked whether metabolite pairs that became dysregulated (i.e., lost correlation with disease) at the first time point could be used to identify individuals that would develop metabolic syndrome at a later time point. To do so, we performed PCA on the 34 metabolites that displayed the strongest evidence for decoherence at the first time point, and used PC1 to ask whether individuals that developed metabolic syndrome at the last time point already showed diagnostic changes (see Materials and methods). Strikingly, we found that PC1, constructed only using data from the first time point, could distinguish between individuals that remained healthy across all time points and individuals that were healthy at the first time point but developed metabolic syndrome later on ($R^2$ = 0.056, $p=3.03\times10^{-10}$, AIC = −448.40; *Figure 3*). This effect was stronger than

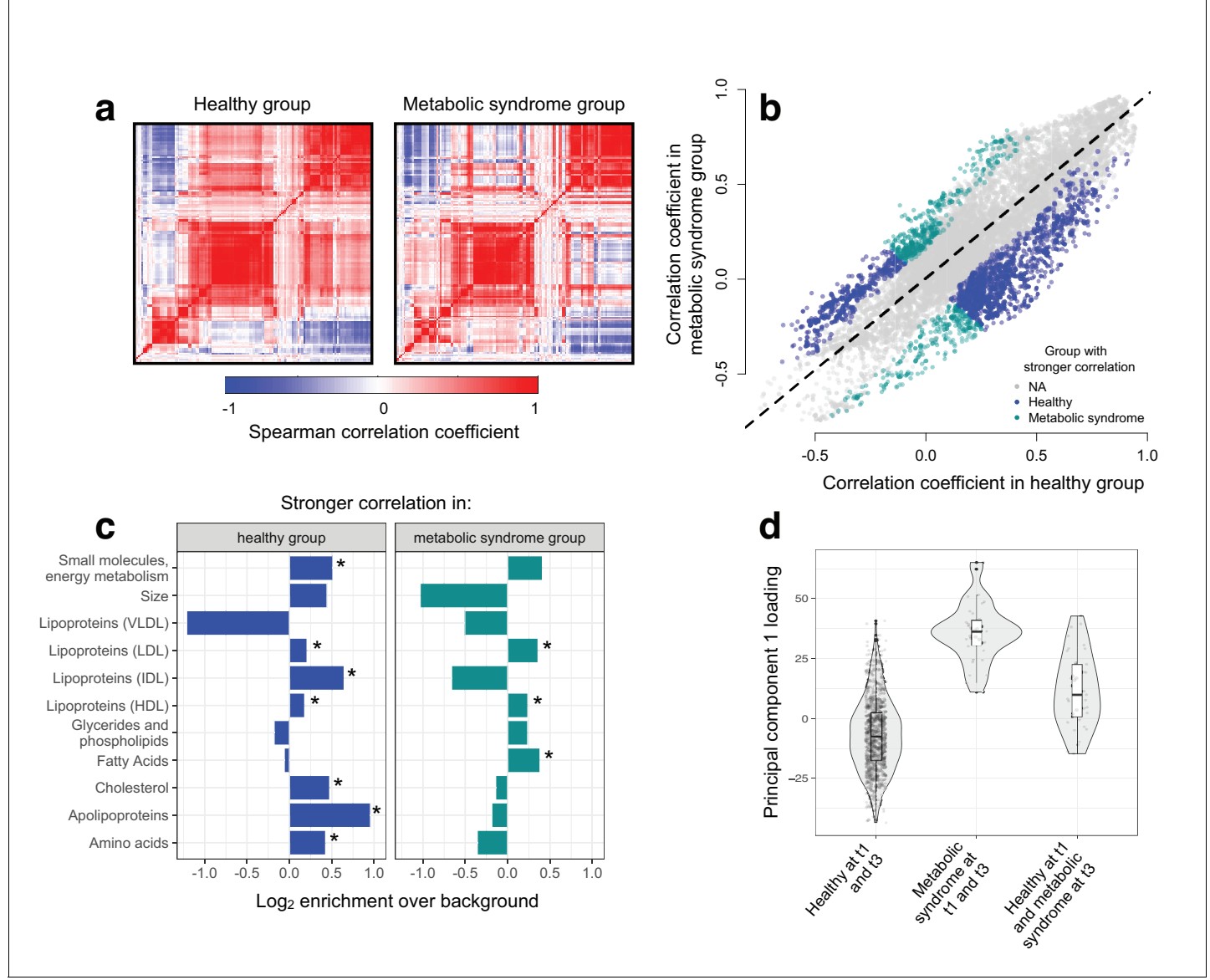

**Figure 3.** Metabolic syndrome leads to decoherence among particular metabolite pairs. (**a**) Correlation matrices showing the magnitude of the Spearman correlation coefficient, for healthy individuals and those with metabolic syndrome. (**b**) Comparison of the Spearman correlation coefficient estimated in healthy individuals (x-axis) versus those with metabolic syndrome (y-axis). Overall, the magnitude of the correlation is similar between groups ($R^2 = 0.62$, $p<10^{-16}$), though the slope of the line is already less than 1, indicating a tendency toward stronger correlations in healthy individuals (beta = 0.814). For the subset of metabolite pairs that display significantly stronger correlations in the healthy (blue dots) or metabolic syndrome class (green dots), this effect is more pronounced (beta = 0.597). (**c**) Categorical enrichment of metabolites that exhibit stronger pairwise correlations in the healthy or metabolic syndrome class (x-axis: $\log_2$ odds ratio from a Fisher's exact test; y-axis: 11 functional classes tested (annotations taken from (***Nath et al., 2017***)). Asterisks indicate significant enrichment. (**d**) A composite measure of metabolites that are strongly decoherent at the first time point (i.e., that show decreases in correlation in individuals with metabolic syndrome) can predict an individual's future health status. Y-axis: Principal component 1 of 34 metabolites, measured at the first time point, with the strongest evidence for dysregulation. X-axis: values are stratified by whether an individual was healthy at the first (t0) and the last (t3) time point, had metabolic syndrome at t0 and t3, or developed metabolic syndrome between t0 and t3 (linear model, $p=3.03\times10^{-10}$).

DOI: https://doi.org/10.7554/eLife.40538.008

The following figure supplement is available for figure 3:

**Figure supplement 1.** QQ-plot confirming that empirical null distributions approximate a uniform distribution.
DOI: https://doi.org/10.7554/eLife.40538.009

that observed in parallel analyses using triglyceride levels at the first time point (a classic biomarker of metabolic syndrome; $R^2 = 0.046$, p=$6.32\times10^{-9}$, AIC = -432.61) or an index created from the 34 metabolites that exhibited the strongest mean differences between healthy and metabolic syndrome individuals at the first time point ($R^2 = 0.037$, p=$1.42\times10^{-6}$, AIC = −422.08).

## Genetic variation impacts co-expression of metabolism-related genes

### Detection and replication of hundreds of correlation QTLs

Finally, we used CILP to identify genetic variants that control the degree of correlation between a pair of gene transcripts, a pattern we refer to as 'correlation QTLs'. Genotypic effects on co-expression could arise through several possible mechanisms. For example, a SNP that disrupts a transcription factor (TF) binding site in a promoter would lead to low levels of co-expression between the TF and the target gene, but only for individuals carrying the disrupting variant. In these individuals, increased expression of the TF would fail to increase expression of the target gene, leading to an association between SNP genotype and variation in co-expression. This is one mechanistic scenario that has been repeatedly proposed to generate variation in co-expression (*de la Fuente, 2010*; *Gaiteri et al., 2014*), with some empirical support (*Nath et al., 2017*). However, the relationship between genetic variation and co-expression is largely unexplored, suggesting that alterative mechanisms may exist that have yet to be uncovered.

To map SNPs that affect the magnitude or direction of pairwise gene expression correlations, we applied CILP to genotype and whole blood-derived gene expression data from the Netherlands Study of Depression and Anxiety (abbreviated 'NESDA' (*Penninx et al., 2008*)). We used a filtered set of 93,197 SNP genotypes and 33,302 gene expression measurements collected for 2477 individuals in our discovery dataset and 1337 individuals in our replication dataset (see Materials and methods). Because our dataset was not well-powered to test all possible pairwise combinations of gene expression measurements against all SNPs, and because we were interested in understanding co-expression patterns among genes important to metabolic diseases, we focused on the 475 probes most strongly associated with BMI. In total, we identified 484 associations between a SNP and variation in co-expression in our NESDA discovery dataset (at a 10% FDR threshold, corresponding to p<$4.6\times10^{-9}$; linear model controlling for sex, age, smoking behavior, major depressive disorder, red blood cell counts, year of sample collection, study phase, and the first five principal components from a PCA on the filtered genotype call set). These 484 associations involved 247 unique probe sets and 51 genotyped SNP, each of which was involved in 1–424 (mean ±s.d. =3.91 ± 26.9) and 1–173 (mean ±s.d.=9.43 ± 34.1) correlation QTLs, respectively (*Figure 4*; *Supplementary file 4*).

To confirm our results, and overcome any potential biases raised by sub-selection of the data, we replicated our correlation QTL in a separate set of NESDA participants. Here, we found that 304/484 correlation QTLs replicated (at a Bonferroni corrected p=$1.03\times10^{-4}$, n = 1337; 428/484 correlation QTLs replicated at a 10% FDR; *Figure 4*). Further, in whole-blood derived gene expression data from 1414 YFS participants, we found that 47/74 testable correlation QTLs replicated at a 10% FDR (0 replicated at a Bonferroni corrected p=$6.76\times10^{-4}$, but the direction of the effects consistently agreed across datasets, binomial test, p<$10^{-16}$; see Materials and methods).

### General characteristics of correlation QTLs

For the list of 484 correlation QTLs we identified, we performed several follow-up analyses to gain mechanistic insight. First, we asked about the physical location of correlation QTLs relative to the genes for which they altered correlation structure (e.g., are the correlation QTLs acting in *cis* or *trans*), as well as the degree to which correlations QTLs also had mean or variance effects on gene expression (*Supplementary file 5*; *Figure 4—figure supplement 1*). Here, we found that 73.76% of correlation QTL were located on the same chromosome as one of the two correlated genes, 14.57% were located on the same chromosome as both correlated genes, and 11.57% were acting in *trans* on both genes. Further, 87.81% of significant correlation QTL were also *cis* eQTL or varQTL for one of the correlated genes, but no correlation QTL had mean or variance effects on both genes (*Figure 4—figure supplement 1*).

Next, we asked whether the set of genes involved in significant correlation QTLs were enriched for (i) particular biological processes and pathways (using Gene Ontology annotations (*Harris et al.,*

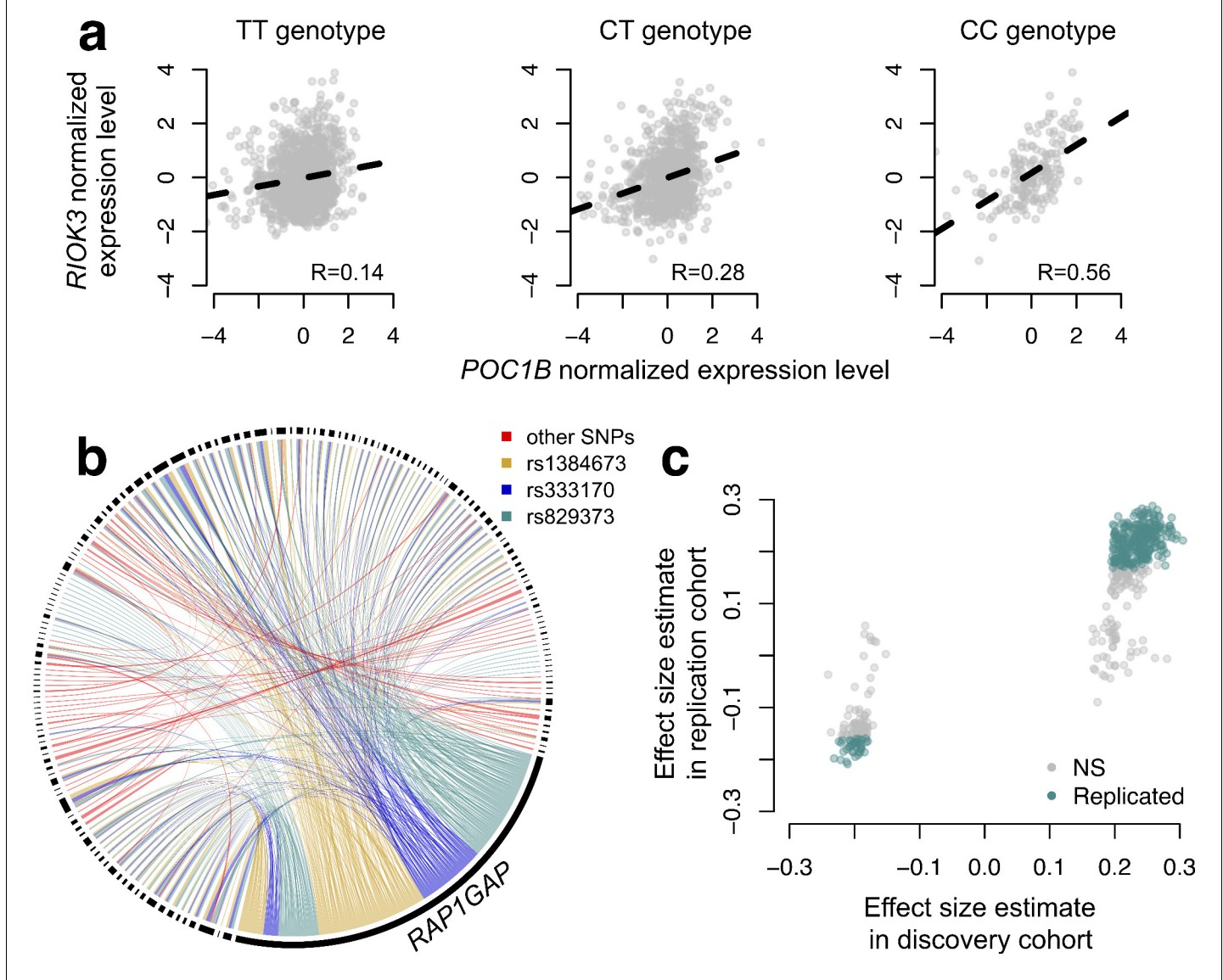

**Figure 4.** CILP approach reveals hundreds of correlation QTLs. (**a**) Example of a correlation QTL, where the SNP rs10953329 controls the magnitude of the correlation between the mRNA expression levels of *POC1B* and *RIOK3*. (**b**) Gene pairs involved in significant (FDR < 10%) correlation QTL. Each black segment represents a gene, and each line connecting two segments represents a significant correlation QTL. Lines are colored by the identity of the SNP controlling the magnitude of the correlation between the gene pair. (**c**) Many correlation QTL identified in the NESDA discovery set (n = 2477) replicate in a second set of NESDA participants (n = 1337). Plot shows effect sizes for each correlation QTL, estimated in the discovery or replication cohort (effect sizes are derived from in *matrixEQTL* (*Shabalin, 2012*)). Points are colored to indicate whether a given correlation QTL passed Bonferroni correction in the replication dataset.

DOI: https://doi.org/10.7554/eLife.40538.010

The following figure supplements are available for figure 4:

**Figure supplement 1.** (**A**) Mean and (**B**) variance effects of SNPs identified as significant correlation QTL.
DOI: https://doi.org/10.7554/eLife.40538.011

**Figure supplement 2.** SNPs driving correlation QTL (between *RAP1GAP* and other many other genes) are also strong cis eQTL for *RAP1GAP*.
DOI: https://doi.org/10.7554/eLife.40538.012

**Figure supplement 3.** Mechanistic relationship between *RAP1GAP* eQTL and correlation QTL.
DOI: https://doi.org/10.7554/eLife.40538.013

**Figure supplement 4.** Comparison of (**A**) p-values and (**B**) effect size estimates from different approaches for identifying correlation QTL.
DOI: https://doi.org/10.7554/eLife.40538.014

**Figure supplement 5.** Cell type effects on (**A**) BMI-gene expression correlations and (**B**) estimations of the correlation QTL effect.

*Figure 4 continued on next page*

*Figure 4 continued*

DOI: https://doi.org/10.7554/eLife.40538.015

**Figure supplement 6.** Genotypes at correlation QTLs do not predict cell type heterogeneity.

DOI: https://doi.org/10.7554/eLife.40538.016

*2004*)), (ii) known TFs, or (iii) known targets of TFs. For analysis (ii) and (iii), we drew on a public database of known TF-target gene interactions created through sentence-based text-mining followed by manual curation (*Han et al., 2015*). We strongly expected genes involved in correlation QTLs to be enriched for TFs or known targets of TFs, given that the mechanisms that have been proposed to generate genetic effects on variation in correlation almost universally involve genotype-dependent TF activity or disruption of TF binding sites (*de la Fuente, 2010*; *Gaiteri et al., 2014*). In support of these ideas, genes involved in significant correlation QTLs were 1.90x more likely to be TFs (hypergeometric test, $p=1.7\times10^{-3}$) and 1.52x more likely to be known targets of transcription factors ($p=6.7\times10^{-4}$) relative to the background set of all genes tested. Additionally, genes involved in significant correlation QTLs were enriched for biological processes involved in metabolism and metabolic disease (*James et al., 2012*; *Rani et al., 2016*) such as cellular response to oxidative stress (hypergeometric test, odds ratio = 2.21, $p<10^{-16}$), intracellular signal transduction ($\log_2$ odds ratio = 1.44, $p<10^{-16}$), and mitophagy (the selective degradation of mitochondria; $\log_2$ odds ratio = 0.82, p=0.018; *Supplementary file 6*).

## Potential mechanisms generating correlation QTL

Many (87.81%) of the correlation QTL we uncovered involve three SNPs (rs1384673, rs333170, and rs829373), which have strong effects on the mean expression of one gene, *RAP1GAP* (*Figure 4—figure supplement 2*). The function of *RAP1GAP* is to convert the transcription factor *RAP1* from its active GTP-bound form to its inactive GDP-bound form, and genes involved in correlation QTL with *RAP1GAP* tend to be closer to *RAP1* binding sites (two-sided Wilcoxon-signed rank test, p=0.04) and to be bound *in vivo* by *RAP1* (Fisher's exact test, odds ratio = 2.14, p=0.03; ChIP-seq data are from mouse embryonic fibroblasts (*Martinez et al., 2010*)). We hypothesize that the *RAP1GAP* correlation QTL 'hotspot' is thus generated because genotype controls whether *RAP1* and its targets are constitutively 'on' or 'off', and variation in *RAP1* activity levels only modulates the expression of *RAP1* target genes within the 'on' genotype (see schematic in *Figure 4—figure supplement 3*). In addition to uncovering a mechanism that can generate large-scale effects on correlation – specifically, cases where the targets of a given TF are constitutively repressed in one genotypic class, but dynamically regulated in another – this example is of interest because previous work has found evidence for converge of *trans* eQTL effects on *RAP1GAP* in whole blood (*Fehrmann et al., 2011*), and has also implicated *RAP1* in the development of insulin resistance, inflammation, and obesity (*Kaneko et al., 2016*).

Though the correlation QTLs we uncovered near *RAP1GAP* were the only 'hotspot' (*Figure 4*), we also found a number of other intriguing examples. For SNPs that regulated the co-expression of a given gene pair in *trans*, we found scenarios such as: (i) regulation of co-expression between genes annotated for the same GO terms (e.g., *GLRX5* and *RPS29* are both annotated for 'metabolic processes'); (ii) regulation of co-expression between a TF and its known target gene (e.g., *GATA2* and *RAB24*); and (iii) regulation of co-expression between disease-associated genes (e.g., *SLC4A1* and *TPM1* are associated with renal (*Alper, 2003*; *Yenchitsomanus et al., 2005*) and cardiovascular disease (*England et al., 2017*), respectively). We note that in these cases the mechanism that induces a correlation QTL is unclear; nevertheless, our correlation QTL screen uncovers novel loci regulating the relationship between disease and metabolic genes that could be the focus of future follow-up work.

## Discussion

Patterns of transcriptional correlation are widely considered to arise from co-regulation between genes. The analysis of co-expression has become an essential tool for the functional interpretation of transcriptional variation (*de la Fuente, 2010*; *Gaiteri et al., 2014*), with increasing relevance for medical diagnosis (*van Dam et al., 2017*; *Vidal et al., 2011*; *Barabási et al., 2011*) and our

understanding of trait evolution (*Wang et al., 2013*; *Sun et al., 2008*; *Southworth et al., 2009*; *Fukushima, 2013*; *Huang et al., 2015*). However, we still have a primitive understanding of the factors that shape correlations between genes. Specifically, how do environmental perturbations alter essential patterns of co-regulation? And, to what degree are some genotypes better than others at buffering these disruptive effects? To address these questions, we developed a novel, generalizable approach to test whether any predictor variable (e.g., environment, genotype, or another variable of interest) affects the degree of correlation between a pair of measured variables. This simple approach, which relies on calculating the product between two variables after normalization, allows us to produce an individual level estimate of correlation rather than a summary statistic for a population (*Figure 1*). Consequently, we are able to study correlation as a *bona fide* trait, and leverage the flexibility of statistical linear modeling approaches to identify factors associated with variation in correlation between individuals.

We begin by investigating the influence of two environmental perturbations on molecular co-regulation. First, we use data from a controlled experiment, where human monocytes were challenged with a simulated bacterial infection (*Fairfax et al., 2014*). Second, we draw on a dataset of 159 metabolites collected from a population-based, longitudinal study of Finnish individuals (*Raitakari et al., 2008*), and contrast correlation patterns between healthy individuals versus those suffering from metabolic syndrome. In both instances, we find that stressful environmental exposures (infection and disease) lead to decoherence, manifested as a widespread decrease in the magnitude of pairwise correlation coefficients between mRNA transcripts or metabolites (*Figure 3*).

Although we expect the relationship between mRNA transcripts and metabolites to change following an environmental perturbation (e.g., both LPS stimulation and metabolic syndrome clearly affect mean gene expression or metabolite levels (*Fairfax et al., 2014*; *Nath et al., 2017*)), the consistent direction of effects we observe on the correlation is striking. Specifically, of all metabolite and mRNA transcript pairs that are significantly differentially correlated, there is a strong directional bias toward a decrease in correlation following the environmental perturbation: 61–74% of significant pairs follow this pattern. In other words, under stress, some genes and metabolites that are typically co-regulated no longer are. Similar biased loses in correlation have also been observed in miRNA pairs measured in the plasma of patients suffering from cognitive impairment versus healthy controls (*Kayano et al., 2016*), as well as in gene expression data collected from aging versus young mice (*Southworth et al., 2009*) and in a wide range of cancers (*Anglani et al., 2014*). However, as this pattern has generally not been tested for explicitly, the degree to which it commonly characterizes disease, aging, or environmental perturbations remains to be seen.

The loss of transcriptional robustness we observe is an intuitive extension of decanalization models (*Gibson, 2009b*). Decanalization models posit that, through many generations of stabilizing selection, biological systems evolve to maintain homeostasis under a certain range of environmental (or genetic) perturbations, and changing the environment beyond this range will result in homeostatic breakdown (*Gibson, 2009a*; *Gibson, 2009b*). This 'breakdown' has traditionally been thought of as an increase in variance (*Geiler-Samerotte et al., 2018*; *Gibson and Wagner, 2000*), and we note that our discussion of decanalization as a change in correlation is a departure from traditional models. Nevertheless, we believe we are potentially seeing decanalization play out at the molecular level, where formerly co-regulated molecules become dysregulated following a shift in the environment. Importantly, the longitudinal nature of the YFS dataset allowed us to track the health status of the same individuals over time and test the hypothesis that decoherence at the molecular level is associated with disease risk. In support of decanalization models, we found that metabolite pairs that lose correlation following disease onset at the first-time point could be used to distinguish between perpetually healthy individuals and those that would develop disease later on. This set of metabolites are particularly interesting from a clinical perspective, as they appear to be especially sensitive to the homeostatic breakdown associated with metabolic syndrome, and could thus potentially serve as biomarkers.

In our final set of analyses, we investigated the role of genetic variation in driving variation in co-expression. To date, work on differential co-expression has largely focused on contrasts between cases and controls (*de la Fuente, 2010*; *van Dam et al., 2017*; *Rotival and Petretto, 2014*), between tissues (*Saha et al., 2017*; *Gao et al., 2016*), or between individuals inhabiting different environments (*de la Fuente, 2010*; *Hsu et al., 2015*), while the genetic basis of differential co-expression has received much less attention. Our ability to map correlation QTLs directly using CILP

fills this gap, and reveals that correlation between mRNA transcripts is under genetic control. Our analyses build on three recent studies that came to similar conclusions using more indirect approaches and/or much smaller sample sizes. First, Nath et al. identified correlated groups of mRNA transcripts and metabolites (termed 'modules'), and performed genome-wide scans to associate specific SNPs with variation in each module (*Nath et al., 2017*). In a related analysis, Lukowski et al. demonstrated that mRNA transcripts with evidence for genetic correlations were more likely to be regulated by the same eQTL (*Lukowski et al., 2017*). Finally, van der Wijst recently leveraged single-cell mRNA-seq data from 45 individuals to build personalized co-expression networks, and used these data to identify genetic variants that predicted inter-individual variation in correlation structure (*van der Wijst et al., 2018*). Together, these studies point toward pervasive genetic control of co-expression, and CILP provides the tools to probe this form of regulation even further. In particular, mapping correlation QTLs across different environments, and compiling a more mechanistic picture of how genetic variation affects correlation dynamics (e.g., using ChIP-seq or ATAC-seq data integrated with correlation QTL scans) are major priorities for future work.

Our transcriptome analyses focused on pre-defined gene sets because of our need to limit the search space. In particular, we focused on differentially expressed or BMI-associated genes (in the Fairfax et al (*Fairfax et al., 2014*). and NESDA (*Penninx et al., 2008*) datasets, respectively), because testing for variation in correlation across all genes expressed in a given tissue would require tens of millions of tests. Applying CILP using unbiased screens will require extremely large sample sizes to overcome the multiple testing burden. Importantly, power to detect variation in correlation using CLIP does increase with sample size. This would not be the case with other methods that compare group rather than individual-level variation in correlation (*de la Fuente, 2010*; *van Dam et al., 2017*); for these methods, increasing sample size would increase the precision of the estimated correlation in a given group, but not directly the power to detect differential correlation.

Lastly, we focused on human genomics in this study because of the availability of large datasets with individual-level data. However, our study builds on extensive previous work in model systems investigating variation in co-expression in yeast (*Wang et al., 2013*; *Sun et al., 2008*), mice (*Southworth et al., 2009*), Arabidopsis (*Fukushima, 2013*; *Mine et al., 2018*; *Zhang et al., 2017*), and Drosophila (*Huang et al., 2015*), all of which point to the critical importance of co-regulation in complex trait variation. The framework we outline here could be particularly useful in conjunction with a growing number of multi-parent mapping populations (*Long et al., 2014*; *Iraqi and Collaborative Cross Consortium, 2012*; *de Koning and McIntyre, 2017*; *Cubillos et al., 2013*; *Pascual et al., 2016*), which provide unprecedented statistical power in model systems. In particular, CILP paired with controlled manipulations in model systems could allow researchers to uncover genotype x environment effects on correlation, a problem that has been notoriously difficult to study in humans. Similarly, while here we focused exclusively on molecular traits, our approach could be paired with many additional data types. Moving beyond genetics, CILP could be used to identify the drivers of community level correlations in ecological datasets, tradeoffs (that manifest as negative correlations) between different fitness components or life history traits, and much more. In essence, questions of how and why correlations between molecular or organism-level phenotypes vary is at the heart of many fields, and we anticipate that our approach will thus be widely applicable.

## Materials and methods

### Simulations

To evaluate the robustness and performance of CILP, we simulated data under a wide range of scenarios, focusing on correlation QTL effects (*Table 1*). We first simulated genotypes for a single locus with a major and minor allele, and with a minor allele frequency of 0.5. We then simulated gene expression data across n = 1000 individuals for 10,000 gene pairs with correlated gene expression levels according to the following model:

$$y_g \sim N\left(\begin{matrix} m1 + b1*g \\ m2 + b2*g \end{matrix}\right), \begin{bmatrix} \sigma_1^2 + v1*g & b*g \\ b*g & \sigma_2^2 + v2*g \end{bmatrix}$$

For each genotype *g* (which can take the values 0 (homozygous major), 1 (heterozygous), or 2

(homozygous minor)), we simulated gene expression values for the two focal genes with covariance $b*g$. True positives were simulated with $b$ ranging from 0.05 to 0.5, and null correlation QTLs were simulated with $b = 0$. The two focal genes had mean expression levels of $m1$ and $m2$, respectively. $b1$ and $b2$ model the expression changes with respect to genotype (i.e., the effect size of the eQTL), and $v1$ and $v2$ model changes in variance with respect to genotype (i.e., the effect size of the varQTL).

## The cardiovascular risk in young Finns study (YFS)

### Study subjects

We used phenotype, genotype, gene expression, and NMR metabolite data derived from a previously described study of unrelated Finnish young adults. The YFS is a longitudinal prospective cohort study initiated in 1980, with follow-ups carried out every 3 years. The study was designed to monitor cardiovascular disease in individuals recruited from five regions in Finland: Helsinki, Kuopio, Oulu, Tampere, and Turku (*Raitakari et al., 2008*; *Nuotio et al., 2014*). Our analyses focused on individuals for whom NMR metabolite data were generated from whole blood samples collected at the 2001 (n = 2248), 2007 (n = 2161), and 2011 (n = 2040) follow-ups. Additionally, we replicated our correlation QTL findings from NESDA using paired genotype and whole blood-derived gene expression data available for a subset of individuals sampled at the 2011 follow-up (n = 1414).

### Metabolite data and metabolic syndrome classification

Metabolic measures were collected from whole blood-derived serum samples collected in 2001 (n = 2248), 2007 (n = 2161), and 2011 (n = 2040) using the procedures described in *Soininen et al. (2009)*. We focused on a set of 159 measures (following *Nath et al., 2017*), of which 148 were directly measured and 11 were derived (*Supplementary file 2*). The 148 measures included molecules from 14 lipoprotein subclasses, two apolipoproteins, eight fatty acids, eight glycerides and phospholipids, nine cholesterols, nine amino acids, and ten small molecules (involved in glycolysis, as well as the citric acid and urea cycle). We refer to these metabolic measures as 'metabolites', but note they represent a more heterogenous set. Annotations for each metabolite were derived from (*Nath et al., 2017*) (*Supplementary file 2*). Measurements that failed quality control filters (*Soininen et al., 2009*) were treated as missing, and measurements of zero were set to the minimum detectable value for a given metabolite (*Nath et al., 2017*).

We removed individuals from our analyses that had type I or II diabetes, or who were reported to be on statin medications. For the remaining dataset, we classified individuals as healthy or having a metabolic syndrome-like phenotype at a given time point using a random forests approach implemented in the R package 'party' (*Strobl et al., 2008*; *Hothorn et al., 2006*). We did so because only a subset of the five data types required to diagnose metabolic syndrome (*Grundy et al., 2005*) were available for all samples and time points. Specifically, fasting serum triglyceride levels, high-density lipoprotein (HDL) cholesterol levels, and blood glucose levels were available for all samples, but waist circumference and blood pressure were only available for a subset of 2011 samples (n = 1414; for details on how these traits were measured, see *Raitakari et al., 2008*; *Nuotio et al., 2014*). We therefore trained a random forests classifier on the subset of the 2011 dataset that could be identified as having metabolic syndrome or not based on standard criteria (*Grundy et al., 2005*) and used the trained model to predict health status for the remaining samples. We used the follow features to generate the random forests: triglyceride levels, HDL cholesterol levels, blood glucose levels, BMI, and sex. We removed individuals from the dataset that were not confidently assigned to either class (i.e., individuals for whom the probability of assignment to either class did not exceed 2/3) leaving us with the following sample sizes: n = 1564 in 2001, 1498 in 2007, and 1501 in 2011. Using data from the 2011 time point (where we have all information necessary to diagnose metabolic syndrome following the American Heart Association's criteria (*Grundy et al., 2005*)), we estimate that our healthy class includes 95% true positives and our metabolic syndrome class includes 84% true positives (*Supplementary file 7*).

We also repeated the analyses described in the main text using all individuals that could not be assigned to either the healthy or metabolic syndrome class confidently as a third 'transitional' class. Here, we coded the 'transitional' class as intermediate between healthy and metabolic syndrome individuals. In doing so we found patterns very similar to those described in the main text:

significantly differentially correlated metabolites were biased toward decoherence, and tended to exhibit stronger correlations in healthy individuals (n = 1110 of 1751 significant pairs displayed this pattern; Spearman correlation among -$\log_{10}$ p-values from this analysis versus the main text analysis: 0.114, $p<10^{-16}$). With respect to the metabolite categories we identified as enriched for decoherent metabolites (*Figure 3*), we only uncovered one new, enriched class when we included 'transitional' individuals in our analysis (namely, fatty acids; $p=1.29\times10^{-4}$, $\log_2$ odds ratio = 0.178).

## Genotype and gene expression data

While our analyses of YFS participants focused primarily on metabolite data, we used paired genotype and gene expression data collected from 1414 participants in 2011 to replicate correlation QTL discovered in NESDA. Whole blood was collected from each individual in PAXgene Blood RNA tubes and was used to perform genome-wide genotyping on a custom Illumina Human 670 k BeadChip array and genome-wide mRNA quantification on the Illumina HumanHT-12 v4 Expression BeadChip (information on microarray experimental and quality control procedures are provided in detail elsewhere (*Smith et al., 2010*)).

Genotype data were filtered prior to analysis, and SNPs that met at least one of the following criteria were excluded from downstream analyses: (i) evidence that the SNP was not in Hardy-Weinberg equilibrium ($p<10^{-6}$), (ii) data missing for >5% of all individuals, (iii) minor allele frequency <5%. Gene expression data were log2 transformed prior to analysis and filtered to remove probe sets that overlapped known SNPs, as well as those that measured lowly expressed transcripts. Additional details on genotype and gene expression quality control and filtering procedures are described elsewhere (*Gusev et al., 2016*).

## Testing for effects of metabolic syndrome on variation in correlation

### Implementing CILP

To understand how differences in health status impact correlation structure, we applied CILP to a set of 159 NMR metabolite measures collected across three time points in the YFS. To remove highly correlated metabolites from our dataset, we first computed the Spearman correlation within each year for all pairs of metabolites (n = 12561, equivalent to 159 chose 2), and excluded pairs with rho >0.9 in any given year. For the remaining 11491 metabolite pairs, we computed the product after z-score transforming each metabolite measure, within year and within each set of healthy and metabolic syndrome-like individuals separately (using the 'scale' function in R).

Using the set of products computed for all filtered pairs of metabolites, we constructed linear mixed effects models in the R package 'nlme' (*Pinheiro et al., 2017*). Specifically, we tested the degree to which each set of products was predicted by health status (healthy/metabolic syndrome-like), controlling for follow-up year (as a factor), sex, age, and individual identity as a random effect. We extracted the p-values associated with the health status effect, and corrected for multiple hypothesis testing. Importantly, we also conducted parallel analyses in which we permuted health status (metabolic syndrome/healthy) prior to (i) data normalization, product computation, and linear mixed model analyses or (ii) prior to linear mixed model analyses only; in both cases, our permutation results suggest that the null distribution approximates the expected uniform distribution (*Figure 3—figure supplement 1*).

### Enrichment of differentially correlated metabolite pairs

To understand whether particular classes of metabolites are more likely to increase or decrease in correlation as a function of health status, we used annotations for each metabolite measurement (*Nath et al., 2017*) (*Supplementary file 2*) coupled with hypergeometric tests. Specifically, we asked whether pairs of metabolites significantly affected by health status were more likely to come from each annotation category, relative to the background set of all tested metabolite pairs. We performed these enrichment tests separately for two groups of metabolite pairs, namely (i) those that exhibited stronger correlations in healthy people relative to individuals with metabolic syndrome and (ii) those that showed the opposite pattern. We corrected for multiple hypothesis testing using a Bonferroni approach, and report the results in *Figure 3* and *Supplementary file 3*.

## Predictive power of metabolite pairs that lose correlation following the onset of disease

Because we had access to metabolite data collected from the same individuals across multiple time points (sample sizes: n = 1564 in 2001, 1498 in 2007, and 1501 in 2011), we asked whether decoherent metabolite pairs could be used to identify individuals that would develop metabolic syndrome at a later time point. To do so, we first identified metabolites with the strongest evidence for decoherence. Specifically, we identified 34 metabolites for which > 0.25 of all tested pairs (involving the focal metabolite and any other metabolite) were significantly less correlated in metabolic syndrome individuals relative to healthy individuals at the first time point. We performed PCA on these 34 metabolites, using data from the first time point only, and asked whether PC1 was related to health status at the last time point (using linear models that controlled for age and sex). For comparison, we performed parallel analyses using triglyceride levels at the first time point, or the 34 metabolites with the strongest mean differences between health classes at the first time point, as our biomarkers of interest.

## The Netherlands study of depression and anxiety (NESDA)

### Study subjects and sample collection

Our correlation QTL analyses focused on phenotype, genotype, and gene expression data derived from the Netherlands Study of Depression and Anxiety (NESDA; n = 5339 participants). NESDA is a previously described cohort study designed to investigate the long-term consequences of depressive and anxiety disorders (*Penninx et al., 2008*). Briefly, whole blood was collected in PAXgene Blood RNA tubes from each NESDA participant, and the Affymetrix Genome-Wide Human SNP Array 6.0 and Human Genome U219 Array were used for genotyping and mRNA quantification, respectively. A number of additional health, demographic, and biochemical traits were also recorded for each participant as described in *Wright et al. (2014)*. Complete white blood counts, consisting of lymphocytes, neutrophils, basophils, monocytes, and eosinophils counts were measured for a subset of blood samples.

NESDA contains twin pairs and families discordant for depressive and anxiety disorders, as well as unrelated individuals. To generate a discovery dataset of unrelated individuals, we removed all members of a given family except for one randomly selected member (leaving us with n = 2477 unrelated individuals). For each family, we then randomly assigned one of the remaining individuals to our replication cohort (leaving us with n = 1337 individuals).

### Genotype and gene expression data

Information on genotype and gene expression microarray data generation and quality control are described in detail elsewhere (*Wright et al., 2014*). We filtered the genotype data from our discovery sample set using the following criteria: >5% minor allele frequency (MAF), >5% of individuals exhibit the homozygous minor genotype, and the focal SNP is not in linkage disequilibrium (LD; $r^2$ <0.5) with other genotyped SNPs within 50 kb. LD filtering was performed using the 'snpgdsLDpruning' function in the R package *SNPRelate* (*Zheng et al., 2012*). This filtering left us with 93,197 SNPs for analysis.

Gene expression data were first log2 transformed and filtered to remove probe sets measuring lowly expressed transcripts. Specifically, we calculated the mean expression level in our discovery cohort for every probe set, and excluded probe sets if this value exceeded the maximum value obtained for any control probe set (which should all theoretically have an expression level of 0). Probe sets were also removed if they overlapped a genotyped, polymorphic SNP in the NESDA cohort or a SNP at >5% MAF in whole genome-sequencing data from 769 Dutch individuals (*Francioli and Genome of the Netherlands Consortium, 2014*). To identify the genomic location of each probe set for this purpose, we downloaded all 25 bp probe sequences for the Human Genome U219 array, mapped these sequences with *STAR* (*Dobin et al., 2013*) to the human reference genome (hg19), and extracted a set of genomic location coordinates for each probe set from the alignment file. We performed intersections of these genomic location coordinates with known SNP locations (downloaded from the Affymetrix website) using *bedtools* (*Quinlan and Hall, 2010*).

## Testing for genetic effects on variation in correlation

### Mapping correlation QTLs in NESDA

We used the filtered set of SNPs and expression probes measured in NESDA to identify associations between individual SNPs and variation in correlation between the expression levels of two transcripts (referred to here as 'correlation QTL'). Because our dataset is not well-powered to test all possible pairwise combinations of gene expression probes against all genotyped SNPs, we explored several ways to reduce the search space. Specifically, we attempted analyses focusing on: (i) pairs of known transcription factors and their target genes (*Han et al., 2015*), tested against SNPs within 1 MB of either gene; (ii) pairs of genes known to be involved in the same biological pathway (*Mi et al., 2016*); and (iii) pairs of genes associated with a trait of interest (specifically, BMI, age, or smoking behavior (smoker/non smoker)). In the main text, we report results for analyses that tested for correlation QTL at BMI-associated genes, as this is the approach that produced the strongest signal.

To identify BMI-associated genes, we constructed a linear model for each filtered probe set measured in the NESDA discovery set. Specifically, we tested for an association between log2 transformed expression levels and BMI, controlling for sex, age, smoking behavior (smoker/non smoker), diagnosis with major depressive disorder (yes/no), red blood cell counts, year of sample collection, study phase, and the first five principal components from a PCA on the filtered genotype call set (obtained from the 'prcomp' function in R). We extracted the p-value associated with the BMI effect from each model, and rank ordered probe sets according to this statistic. We retained the top 300 unique genes for downstream analyses, which corresponded to 475 unique probe sets (because multiple probe sets, in most cases tagging different exons, exist on the array for a given gene).

To calculate correlation among the set of 475 gene expression measurements associated with BMI, we used a slightly modified version of the approach described in Results. Specifically, we first removed any mean effects of the following major covariates by running a linear model on the log2 transformed expression values and extracting the residuals: BMI, sex, age, smoking behavior, diagnosis with major depressive disorder, red blood cell counts, year of sample collection, and study phase. We further normalized the residuals for each probe set to mean 0 and unit variance using the 'scale' function in R, and computed the product of all possible pairwise probe set combinations (n = 112,575 combinations, equivalent to 475 chose 2). We used this matrix of products as the input for our correlation QTL screen. For our screen, we relied on *matrixEQTL* (*Shabalin, 2012*) to test for an association between each SNP passing filters (n = 93,197) and each vector of products, controlling for BMI, sex, age, smoking behavior, diagnosis with major depressive disorder, and the first five principal components from a PCA on the filtered genotype call set.

### Annotation and analyses of correlation QTLs

We identified 484 associations between a given SNP and variation in correlation between two probe sets in our NESDA discovery dataset (at a 10% FDR threshold (*Benjamini and Hochberg, 1995*)). For this list of 484 associations, we performed several follow-up analyses. First, we asked whether each correlation QTL acted as a mean or variance QTL for either of the two co-expressed genes (*Figure 4—figure supplement 1*). To do so, we used gene expression data that had been normalized and covariate corrected (for year of sample collection, study phase, and red blood cell content). To test for eQTL, we used linear models controlling for BMI, sex, age, smoking behavior, and major depressive disorder. To test for variance QTL, we used the R package 'dglm' (*Dunn and Smyth, 2016*) and controlled for the same covariates.

Second, we asked whether the set of genes involved in significant correlation QTLs were enriched for (i) known transcription factors or (ii) known targets of transcription factors compared to the background set of all genes tested for correlation QTLs. To do so, we used a database of transcription factor-gene target associations and hypergeometric tests (*Han et al., 2015*).

Third, we asked whether the set of genes involved in significant correlation QTLs were enriched for specific gene ontology categories (*Harris et al., 2004*) compared to the background set of all 300 genes tested (*Supplementary file 6*). To do so, we used the R package 'mygene' and hypergeometric tests.

Fourth, many of the correlation QTLs involved one gene, *RAP1GAP*, whose primary function is to switch the transcription factor *RAP1* from its active GTP-bound form to its inactive GDP-bound form. Therefore, we asked whether genes involved in correlation QTLs with *RAP1GAP* were more likely to

be bound by *RAP1*, using ChIP-seq data from mouse embryonic fibroblasts (*Martinez et al., 2010*). To do so, we lifted over coordinates for 30,398 *RAP1* binding sites from the mouse genome (mm9) to the human genome (hg19) using the UCSC Genome Browser *liftOver* tool (*Karolchik et al., 2014*). We then tested (using Fisher's Exact tests) whether genes involved in correlation QTL with *RAP1GAP* were more likely to be bound by *RAP1*, compared to genes not involved in any significant correlation QTL (n = 139 genes). We also assigned each gene to its nearest *RAP1* ChIP-seq peak, and tested whether the absolute distance to the nearest peak (from the gene's annotated transcription start or end site (*Karolchik et al., 2014*)) significantly differed for either of the two comparisons sets described above. If the nearest *RAP1* ChIP-seq peak was within the gene, the distance was coded as 0. We used Wilcoxon signed-rank tests to perform these comparisons. We note that embryonic fibroblasts are not the ideal tissue for understanding biological processes in blood, but *RAP1* is expressed ubiquitously across almost all major tissues (*Ardlie and GTEx Consortium, 2015*) and its binding behavior may thus be conserved.

## Replication of correlation QTLs in NESDA and YFS

For the list of 484 correlation QTLs identified in our NESDA discovery set, we performed association tests in the NESDA replication group (n = 1337) following the same procedures described in *Mapping correlation QTLs in NESDA*.

For each of the 484 correlation QTLs, we also implemented parallel association testing in the YFS (n = 1414) for probe-SNP combinations that met the following criteria: (i) probes that measured both of the focal genes passed all quality control and expression level filters in YFS and (ii) the focal correlation SNP was also typed in YFS. This filtering criteria left us with 74 associations to potentially replicate in YFS. To do so, we first removed mean effects of BMI, sex, age, smoking behavior, and sampling location using linear models applied to log2 transformed expression values. We normalized these residuals using the 'scale' function in R, and computed the product for each of the 74 focal probe pairs. We used these products to test for an effect of each focal SNP, controlling for BMI, sex, age, smoking behavior, and the first five principal components from a PCA on the filtered genotype call set.

## Comparison of approaches for quantifying correlation

Our analyses of infection and metabolic syndrome effects on correlation focused on data that were z-score normalized within each sample group (as described in Results). The parallel of this approach for mapping correlation QTLs would be to z-score normalize each transcript within each genotypic class before computing the product between two focal transcripts. Such an approach has the advantage of removing any mean or variance effects of the predictor variable on the outcome variables. However, this specific pipeline is infeasible for genome-wide QTL mapping, as it would require us to recalculate the outcome variable for every association test we performed (i.e., we would need to re-normalize each transcript pair and re-calculate the product every time we tested a new SNP for an association with correlation). This would be computationally costly, and would preclude us from using efficient tools for QTL mapping (*Shabalin, 2012*).

As an alternative, we therefore regressed out mean effects of major covariates from our gene expression data and normalized the resulting residuals before computing the product between two transcripts. This approach is different than normalizing each transcript within each genotypic class, but attempts to circumvent the same set of potential issues and has the advantage of only needing to be performed once. For comparison, we present the effect size estimates and p-values from the two approaches for the set of 484 significant correlation QTLs we identified (*Figure 4—figure supplement 4*). We also redid the simulations presented in *Table 1* using this alternative approach (*Supplementary file 1*); we note that with this version of CILP, SNPs that are eQTL or proportion QTL for both transcripts can produce false positive correlation QTL. However, we do not find any evidence that this pattern explains our observed results (*Figure 4—figure supplement 1*).

## Cell type heterogeneity-related confounds

### Overview of potential issues

Our correlation QTL analyses focused on gene expression levels measured in whole blood, which is composed of many cell types with distinct transcription profiles. All of our analyses controlled for

one measure of cell type heterogeneity – the proportion of red blood cells in a given sample – however this crude measure may not account for all possible issues raised by cell type heterogeneity. In particular, there are three potential concerns:

1. The mean or variance of whole blood gene expression levels are affected by cell type heterogeneity.
2. BMI is associated with the proportions of particular cell types, such that the BMI-associated genes we tested for correlation QTLs are actually associated with cell type composition (and not BMI).
3. False positive correlation QTLs are produced by genetic effects on cell type heterogeneity. Specifically, if genotype affects cell type proportion, and the pair of genes tested for correlation QTLs are expressed in a cell type specific manner, then individuals of a given genotype would have high expression levels of the two focal genes (and thus high values of their product) because of variation in cell composition, rather than genotypic effects on correlation.

## Cell type proportion effects on whole blood gene expression

Intuitively, we expect potential issue (i) to increase noise in the data, but not to produce false positives (because cell type effects are not connected to genotype). This scenario is equivalent to any other process that generates mean or variance differences in gene expression levels that are orthogonal to genotype, such as environmental or batch effects. Our simulations confirm this intuition and show that the false positive rate does not exceed the expected 5% in the presence of increases in the mean or variance of expression levels (*Table 1*).

Additionally, we inferred 22 measures of cell type heterogeneity (listed in *Supplementary file 8*), and asked whether the correlation QTL results presented in the main text changed when cell type was regressed out of whole blood gene expression data prior to correlation QTL testing. To infer cell type proportions, we performed deconvolution by applying the program CIBERSORT (*Newman et al., 2015*) to normalized gene expression data that had been corrected for batch effects (specifically, we regressed out year of sample collection and study phase and kept the residuals). We checked the accuracy of the deconvolution by comparing empirical and inferred estimates of the proportion of monocytes, eosinophils, and neutrophils (empirical measures of these 3 cell types were available for a subset of NESDA participants, n = 594). Across all 3 cell types, the empirical and inferred measures were highly correlated (Pearson correlations; monocytes: cor = 0.287, p=$9.757\times10^{-13}$, eosinophils: cor = 0.655, p<$10^{-16}$, neutrophils: cor = 0.652, p<$10^{-16}$). Finally, when we compared the p-values for the correlation effect obtained from the pipeline described in the main text, versus a pipeline that regressed out 22 measures of cell type heterogeneity from the gene expression data, the results were extremely similar (Pearson correlation, cor = 0.937, p<$10^{-16}$; *Figure 4—figure supplement 5*).

## Cell type proportion associations with BMI

Potential issue (ii) would also not produce false positive correlation QTL, but would complicate the biological interpretation of our results if we did not robustly identify BMI-associated genes. To explore this possibility, we gathered the dataset described in *Supplementary file 8*, which includes: seven empirical cell type proportion measures available for a subset of or all NESDA participants, 22 measures inferred through deconvolution as described above, and one measure inferred through PCA that is thought to be a proxy for reticulocyte count and has been previously associated with BMI (*Preininger et al., 2013*; *Wingo and Gibson, 2015*).

We used these 7 empirical and 23 inferred measures of cell type heterogeneity to test for an association between each measure of cell type composition and BMI, controlling for covariates (namely, sex, age, smoking behavior, diagnosis with major depressive disorder, year of sample collection, and study phase). BMI was associated most strongly with the PCA-based measure and with red blood cell counts, which was included as a covariate in all models presented in the main text (linear model, both p<$10^{-16}$; full results are presented in *Supplementary file 8*). Importantly, however, these cell type associations with BMI generally did not change the relationship between BMI and gene expression among the set of genes tested for correlation QTLs: 84.2% of the transcripts we identified as BMI-associated in the absence of fine-grained cell-type data are still significantly associated with BMI when these covariates are included (*Figure 4—figure supplement 5*).

## Genetic effects on cell type proportions

Finally, we used our set of 7 empirical and 23 inferred measures of cell type proportion to confirm that the correlation QTLs we identified are not in fact 'proportion QTLs' (i.e., we confirmed that SNPs that affect correlation do not also affect cell type proportion, which could produce false positives under scenario (iii); see *Table 1*). Across the 484 correlation QTLs we identify, we observed little evidence that genotype predicted cell type heterogeneity (using linear models that controlled for sex, age, smoking behavior, diagnosis with major depressive disorder, year of sample collection, and study phase; *Figure 4—figure supplement 6*). Only 2 SNPs were significantly (FDR < 10%) associated with any measures of cell type proportion (rs139170 affects the proportion of plasma cells, $p=3.38\times10^{-4}$; rs2969125 affects the proportion of naïve CD4 +T cells, $p=2.21\times10^{-5}$). Importantly, in neither case are the expression levels of both transcripts also associated with cell type proportion, a condition that would be required to produce a false positive correlation QTL (*Table 1*; *Supplementary file 9*). Thus, we conclude that the correlation QTLs we identified are likely not artifacts of cell type heterogeneity.

## Data and code availability

The NESDA dataset is under controlled access due to confidentiality issues related to the handling of human subjects data, as indicated in the NIH Genomic Data Sharing (GDS) Policy. Researchers can request access at the following website: https://www.ncbi.nlm.nih.gov/projects/gap/cgi-bin/study.cgi?study_id=phs000486.v1.p1. The YFS dataset also comprises health-related participant data and their use is restricted under the regulations on professional secrecy (Act on the Openness of Government Activities, 612/1999) and on sensitive personal data (Personal Data Act, 523/1999, implementing the EU data protection directive 95/46/EC). Due to these legal restrictions, the Ethics Committee of the Hospital District of Southwest Finland has stated that individual level data cannot be stored in public repositories or otherwise made publicly available. However, sharing of YFS data with the scientific community is routinely done via collaboration, following signing of a data-sharing agreement. Investigators can submit an expression of interest to the chairmen of the publication committee (Olli Raitakari: olli.raitakari@utu.fi or Terho Lehtimäki: Terho-Lehtimaki@uta.fi) regarding the submission of a data sharing agreement.

Code for the simulations described here and for implementing CILP can be found at https://github.com/AmandaJLea/differential_correlation (*Lea, 2019*; copy archived at https://github.com/elifesciences-publications/differential_correlation).

## Acknowledgments

We thank all volunteers who participated in the studies described here. We also thank the Ayroles lab for helpful comments on previous manuscript versions. This study was funded by National Institutes of Health (NIH) grants GM124881 to JFA, HL-095056 and HL-28481. AJL is supported by a postdoctoral fellowship from the Helen Hay Whitney Foundation, and AK is supported by NIH grant F31HL127921. MAK is supported by a Senior Research Fellowship from the National Health and Medical Research Council (NHMRC) of Australia (APP1158958). He also works in a unit that is supported by the University of Bristol and UK Medical Research Council (MC_UU_12013/1). The Baker Institute is supported in part by the Victorian Government's Operational Infrastructure Support Program. The Young Finns Study has been financially supported by the Academy of Finland: grants 286284, 134309 (Eye), 126925, 121584, 124282, 129378 (Salve), 117787 (Gendi), and 41071 (Skidi); the Social Insurance Institution of Finland; Competitive State Research Financing of the Expert Responsibility area of Kuopio, Tampere and Turku University Hospitals (grant X51001); Juho Vainio Foundation; Paavo Nurmi Foundation; Finnish Foundation for Cardiovascular Research; Finnish Cultural Foundation; The Sigrid Juselius Foundation; Tampere Tuberculosis Foundation; Emil Aaltonen Foundation; Yrjö Jahnsson Foundation; Signe and Ane Gyllenberg Foundation; Diabetes Research Foundation of Finnish Diabetes Association; EU Horizon 2020 (grant 755320 for TAXINOMISIS); European Research Council (grant 742927 for MULTIEPIGEN project); and Tampere University Hospital Supporting Foundation.

## Additional information

### Funding

| Funder | Grant reference number | Author |
|---|---|---|
| Helen Hay Whitney Foundation | Postdoctoral Fellowship | Amanda Lea |
| National Institute of General Medical Sciences | F31HL127921 | Arthur Ko |
| European Research Council | 742927 | Terho Lehtimäki<br>Emma Raitoharju<br>Mika Kähönen<br>Ilkka Seppälä<br>Nina Mononen<br>Olli T Raitakari |
| Suomen Akatemia | 286284 | Terho Lehtimäki<br>Emma Raitoharju<br>Mika Kähönen<br>Ilkka Seppälä<br>Nina Mononen<br>Olli T Raitakari |
| Horizon 2020 Framework Programme | 755320 | Terho Lehtimäki<br>Emma Raitoharju<br>Mika Kähönen<br>Ilkka Seppälä<br>Nina Mononen<br>Olli T Raitakari |
| Suomen Akatemia | 134309 (Eye) | Terho Lehtimäki<br>Emma Raitoharju<br>Mika Kähönen<br>Ilkka Seppälä<br>Nina Mononen<br>Olli T Raitakari |
| Suomen Akatemia | 126925 | Terho Lehtimäki<br>Emma Raitoharju<br>Mika Kähönen<br>Ilkka Seppälä<br>Nina Mononen<br>Olli T Raitakari |
| Suomen Akatemia | 121584 | Terho Lehtimäki<br>Emma Raitoharju<br>Mika Kähönen<br>Ilkka Seppälä<br>Nina Mononen<br>Olli T Raitakari |
| Suomen Akatemia | 124282 | Terho Lehtimäki<br>Emma Raitoharju<br>Mika Kähönen<br>Ilkka Seppälä<br>Nina Mononen<br>Olli T Raitakari |
| Suomen Akatemia | 129378 (Salve) | Terho Lehtimäki<br>Emma Raitoharju<br>Mika Kähönen<br>Ilkka Seppälä<br>Nina Mononen<br>Olli T Raitakari |
| Suomen Akatemia | 117787 (Gendi) | Terho Lehtimäki<br>Emma Raitoharju<br>Mika Kähönen<br>Ilkka Seppälä<br>Nina Mononen<br>Olli T Raitakari |

| | | |
|---|---|---|
| Suomen Akatemia | 41071 (Skidi) | Terho Lehtimäki<br>Emma Raitoharju<br>Mika Kähönen<br>Ilkka Seppälä<br>Nina Mononen<br>Olli T Raitakari |
| The Social Insurance Institution of Finland | | Terho Lehtimäki<br>Emma Raitoharju<br>Mika Kähönen<br>Ilkka Seppälä<br>Nina Mononen<br>Olli T Raitakari |
| Competitive State Research Financing of the Expert Responsibility area of Kuopio, Tampere and Turku University Hospitals | X51001 | Terho Lehtimäki<br>Emma Raitoharju<br>Mika Kähönen<br>Ilkka Seppälä<br>Nina Mononen<br>Olli T Raitakari |
| Juho Vainio Foundation | | Terho Lehtimäki<br>Emma Raitoharju<br>Mika Kähönen<br>Ilkka Seppälä<br>Nina Mononen<br>Olli T Raitakari |
| Paavo Nurmi Foundation | | Terho Lehtimäki<br>Emma Raitoharju<br>Mika Kähönen<br>Ilkka Seppälä<br>Nina Mononen<br>Olli T Raitakari |
| Finnish Foundation for Cardiovascular Research | | Terho Lehtimäki<br>Emma Raitoharju<br>Mika Kähönen<br>Ilkka Seppälä<br>Nina Mononen<br>Olli T Raitakari |
| Finnish Cultural Foundation | | Terho Lehtimäki<br>Emma Raitoharju<br>Mika Kähönen<br>Ilkka Seppälä<br>Nina Mononen<br>Olli T Raitakari |
| Sigrid Jusélius Foundation | | Terho Lehtimäki<br>Emma Raitoharju<br>Mika Kähönen<br>Ilkka Seppälä<br>Nina Mononen<br>Olli T Raitakari<br>Mika Ala-Korpela |
| Tampere Tuberculosis Foundation | | Terho Lehtimäki<br>Emma Raitoharju<br>Mika Kähönen<br>Ilkka Seppälä<br>Nina Mononen<br>Olli T Raitakari |
| Emil Aaltonen Foundation | | Terho Lehtimäki<br>Emma Raitoharju<br>Mika Kähönen<br>Ilkka Seppälä<br>Nina Mononen<br>Olli T Raitakari |
| Yrjö Jahnsson Foundation | | Terho Lehtimäki<br>Emma Raitoharju<br>Mika Kähönen<br>Ilkka Seppälä<br>Nina Mononen<br>Olli T Raitakari |

| | | |
|---|---|---|
| Signe and Ane Gyllenberg Foundation | | Terho Lehtimäki<br>Emma Raitoharju<br>Mika Kähönen<br>Ilkka Seppälä<br>Nina Mononen<br>Olli T Raitakari |
| Tampere University Hospital | | Terho Lehtimäki<br>Emma Raitoharju<br>Mika Kähönen<br>Ilkka Seppälä<br>Nina Mononen<br>Olli T Raitakari |
| National Health and Medical Research Council | APP1158958 | Mika Ala-Korpela |
| Diabetesliitto | | Päivi Pajukanta |
| National Institute of General Medical Sciences | GM124881 | Julien F Ayroles |

The funders had no role in study design, data collection and interpretation, or the decision to submit the work for publication.

## Author contributions

Amanda Lea, Conceptualization, Data curation, Software, Formal analysis, Validation, Investigation, Visualization, Methodology, Writing—original draft, Project administration, Writing—review and editing; Meena Subramaniam, Conceptualization, Data curation, Software, Formal analysis, Validation, Visualization, Methodology, Writing—review and editing; Arthur Ko, Resources, Data curation, Formal analysis, Methodology; Terho Lehtimäki, Emma Raitoharju, Mika Kähönen, Ilkka Seppälä, Nina Mononen, Olli T Raitakari, Mika Ala-Korpela, Resources, Data curation; Päivi Pajukanta, Resources, Supervision, Funding acquisition, Investigation, Writing—review and editing; Noah Zaitlen, Conceptualization, Resources, Data curation, Software, Formal analysis, Supervision, Funding acquisition, Investigation, Visualization, Methodology, Writing—review and editing; Julien F Ayroles, Conceptualization, Data curation, Funding acquisition, Investigation, Methodology, Writing—original draft, Project administration, Writing—review and editing

## Author ORCIDs

Amanda Lea (iD) http://orcid.org/0000-0002-8827-2750
Arthur Ko (iD) http://orcid.org/0000-0002-1523-7225
Julien F Ayroles (iD) https://orcid.org/0000-0001-8729-0511

## Decision letter and Author response

Decision letter https://doi.org/10.7554/eLife.40538.029
Author response https://doi.org/10.7554/eLife.40538.030

# Additional files

## Supplementary files

• Supplementary file 1. Power to detect simulated correlation QTL using a modified version of CILP (n = 1000 for all simulations).
DOI: https://doi.org/10.7554/eLife.40538.017

• Supplementary file 2. List of analyzed metabolites and functional annotations (following *Nath et al., 2017*).
DOI: https://doi.org/10.7554/eLife.40538.018

• Supplementary file 3. Enrichment of metabolite functional annotations among significantly differentially correlated pairs.
DOI: https://doi.org/10.7554/eLife.40538.019

• Supplementary file 4. Significant correlation QTL in NESDA.
DOI: https://doi.org/10.7554/eLife.40538.020

• Supplementary file 5. Location of correlation QTL SNPs in relation to gene expression probes.
DOI: https://doi.org/10.7554/eLife.40538.021

• Supplementary file 6. Enrichment of gene ontology categories among genes involved in significant correlation QTL.
DOI: https://doi.org/10.7554/eLife.40538.022

• Supplementary file 7. Individuals classified as healthy or having metabolic syndrome based on a random forests classifier.
DOI: https://doi.org/10.7554/eLife.40538.023

• Supplementary file 8. Effects of cell type heterogeneity on BMI.
DOI: https://doi.org/10.7554/eLife.40538.024

• Supplementary file 9. Effects of genotype on cell type heterogeneity.
DOI: https://doi.org/10.7554/eLife.40538.025

### Data availability

The NESDA dataset is under controlled access due to confidentiality issues related to the handling of human subjects data, as indicated in the NIH Genomic Data Sharing (GDS) Policy. Researchers can request access at the following website: https://www.ncbi.nlm.nih.gov/projects/gap/cgi-bin/study.cgi?study_id=phs000486.v1.p1. The YFS dataset (http://youngfinnsstudy.utu.fi/) also comprises health-related participant data and their use is restricted under the regulations on professional secrecy (Act on the Openness of Government Activities, 612/1999) and on sensitive personal data (Personal Data Act, 523/1999, implementing the EU data protection directive 95/46/EC). Due to these legal restrictions, the Ethics Committee of the Hospital District of Southwest Finland has stated that individual level data cannot be stored in public repositories or otherwise made publicly available. However, sharing of YFS data with the scientific community is routinely done via collaboration, following signing of a data-sharing agreement. Investigators can submit an expression of interest to the chairmen of the publication committee (Olli Raitakari: olli.raitakari@utu.fi or Terho Lehtimäki: Terho-Lehtimaki@uta.fi) regarding the submission of a data sharing agreement. Code for the simulations described here and for implementing CILP can be found at https://github.com/AmandaJLea/differential_correlation (copy archived at https://github.com/elifesciences-publications/differential_correlation).

The following previously published dataset was used:

| Author(s) | Year | Dataset title | Dataset URL | Database and Identifier |
|---|---|---|---|---|
| Wright FA, Sullivan PF | 2015 | Integration of Genomics and Transcriptomics in unselected Twins and in Major Depression | https://www.ncbi.nlm.nih.gov/projects/gap/cgi-bin/study.cgi?study_id=phs000486.v1.p1 | NCBI database of Genotypes and Phenotypes, phs000486.v1.p1 |

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
