## [Decision Letter]

[**Editorial note:** This article has been through an editorial process in which the authors decide how to respond to the issues raised during peer review. The Reviewing Editor's assessment is that all the issues have been addressed.]

Thank you for submitting your article "Genetic and environmental perturbations lead to regulatory decoherence" for consideration by *eLife*. Your article has been reviewed by two peer reviewers, and the evaluation has been overseen by a Reviewing Editor and Diethard Tautz as the Senior Editor. The following individual involved in review of your submission has agreed to reveal identity: Greg Gibson (Reviewer #1). The other reviewer remains anonymous.

The reviewers, and I, are convinced of the value of your approach (as one reviewer points out, this is long overdue). We do indeed have a limited understanding of the factors that shape correlations between genes and your work addresses this problem. In particular your observation that decoherent metabolites are good predictors is noteworthy, as is your successful identification of 'correlation-QTLs'. Though perhaps it is not surprising that the degree of correlation is under genetic control, it's certainly good to have that confirmed and analysed, as you have done here. Altogether, your paper represents an important advance.

I want to highlight one issue, to do with unacknowledged confounds. As one of the reviewers points out, the inclusion of expression data from multi-cellular tissues like blood is problematic, and I expected you might incorporate into your method one of the more recent analytical approaches to decompose the contributions (e.g. Nat Methods. 2017 Feb 28;14(3):218-219. doi: 10.1038/nmeth.4190). Would that improve the signal?

Separate reviews (please respond to each point):

*Reviewer #1:*

The manuscript on decoherence by Amanda Lee and colleagues presents a novel, and in my view long over-due, framework for considering decay of modularity of gene expression in disease conditions (among other applications). Alternative approaches describing statistical support for conservation of modules do not have resolution to individual transcripts or metabolites, so this approach holds considerable promise. I thought that the Introduction and Discussion sections were particularly insightful. I also applaud *eLife* for the experiment in "acceptance prior to review" and am happy to support this in this case. I hope that two more major sets of comments, followed by some minor ones, are helpful to the authors.

My first comment goes to the transparency of the description of CILP and its relationship to canalization. I think the way it is presented is overly complicated. If I am not mistaken, CILP is simply a t-test (or potentially ANOVA) of the Pearson correlation coefficients for all pairs of biomarkers in an analysis. The formula at the start of the Results section is that quantity, but then you say it is correlate with some random variable x. In actuality x is a fixed effect in all your comparisons, and it would be clearer to call it a test of association with x since it is the biomarkers that are being correlated. Furthermore, in Figure 3 you switch to the Spearman correlation, which I can see being appropriate, but is at odds with the rest of the description. Also to this point, canalization is commonly used in reference to increase in variance, which I can see could lead to de-coherence, but that is neither necessary nor sufficient, and this could be remarked upon.

My second comment has to do with the effect of unaccounted for factors on covariance. All transcriptomes and metabolomes have a high correlation structure, due to a combination of trans-acting influences, environmental influences, and mixtures of cell types. As far as I can see, the opening simulation study does not incorporate overall correlation, it only models pairwise effects due to cis-acting variation and correlation-QTL as defined by the authors. This should be acknowledged: it is extremely hard to model true transcriptional variation, but I suspect it makes a big difference. (Parenthetically, why call the cis effects b1 and b2, and the correlation effects b – another symbol for the latter would also be less confusing). Regarding the blood gene expression datasets, in the Materials and methods you consider the possibility that cell-type mixtures matter but don't find any influence of available cell counts. However, I can tell you from extensive experience that the major impact on whole blood gene expression in relation to BMI is reticulocyte count, which surprisingly is unrelated to RBC count, but if you fit Axis 2 from Preininger et al. (PMID: 23516379), renamed as Axis R in PMID: 29950172 and PMID: 30223463 and shown to strongly associate with BMI in PMID: 25300922 and PMID: 23162571 and countless other studies we've looked at, it is basically reticulocyte count. I certainly am not asking you to reference these studies (!) but do suggest you fit Axis 2 and see if it has an effect, because it is one of the major components of whole blood expression. I agree that the other Axes that correlate with B and T cell or Neutrophils only weakly associate with BMI.

Minor Comments:

Here are a few more minor comments:

Figure 2. Panel (C) described in the Legend has been removed from the version in the manuscript

LPS study: Need to reference the Fairfax et al. dataset, in the text (you don't do so until the Discussion). Also note that they describe a number of eQTL that have significantly different effects at BL, 2hr LPS, and 24hr LPS, some switching between the two exposure times. Do your correlation QTL tend to be the same, or overlap with, those instances?

Metabolic study. It is not clear from the text whether the results in Figure 3A to 3C are from t0 or from a combined analysis. I suspect just t0, please clarify. But I am also concerned that there may be inflation of the prediction of disease progression in Figure 3D due to use of the discovery dataset in the testing phase (winner's curse), especially if you used t1 and t3 in the discovery. (There will in any case be a bias due to correlations across times within individuals). I'd rather see a more robust analysis that uses a subset of the individuals and then predicts expression in the remainder – there should be sufficient sample size for this.

Subsection “Identification of disease-associated variation in metabolite correlation”, instead of stating the percentages of the subset of genes (74.4 and 25.6), state the percentages of all genes, it will still be 3:1.

Is there any indication that the transcripts that encode the enzymes that regulate the de-coherent metabolite sets are also de-coherent? I guess you may need data of both types from the same individuals, or from liver.

Regarding the correlation QTLs in NESDA, the first thing I think you should report is what fraction of them are also main effect cis-eQTL for one or the other transcript in each pair. Concerning RAP1GAP specifically, is it a trans-eQTL hotspot (I have some vague recollection that it may be, in blood). The enrichment statistics for its binding sites do not convince me at all: P<0.03 is a bit of a joke, and fibroblasts are not relevant, I don't think that result helps at all. Regarding the TF enrichment, I think you need to define what you mean by "involved in" since we now know that the locally regulated gene is usually not the closest one.

Finally, it would be interesting to know what happens to the expression in individuals who do not fall into the Healthy or MetS classes. Such ambiguous cases may have even more extreme de-coherence as they are transitional, at least at one of the time points. If the result is interesting it could be worth reporting.

Thanks for the opportunity to review such an interesting manuscript. Good luck with it.

Additional data files and statistical comments:

If the authors address some of the comments above, they will be addressing some rigor and reproducibility issues. I do not see a need for additional files.

*Reviewer #2:*

In this work, the authors start to investigate the potential for using the correlation between genes/metabolites as a distinct trait that may be parsable from the mean and variance of the same components. Overall, the approach seems solid albeit at points the strongest available support does not seem to be being used which does hinder the papers potential broad appeal. It is as if the need to write to clinical geneticists has slightly hamstrung the authors from really diving into the concepts that are actually very interesting and thought provoking. I have some specific questions and changes below.

On a general note, the paper is written in the general where this approach and its inferences are possibly of use in non-human systems. However, there isn't an obvious effort to discuss work in other systems, *Drosophila*, yeast, Arabidopsis, Maize, etc, that might influence the ideas and conclusions of this paper. There is similar work in these systems (See below) that touches on these topics and if the goal is really a broad audience they should be cited. If the goal is just a human audience than every term should be caveated by the word human, i.e. evolutionary genetics should be human evolutionary genetics, biology should be human biology, etc. Human Genetics can benefit from reading and acknowledging the work in other systems especially as Human Genetics greatly benefits from the of excellent young senior authors who come from other systems like *Drosophila*.

In the Introduction, the authors say "asking whether predefined sets of genes are differentially co-expressed between groups. Almost universally, these approaches are only designed for comparisons between two groups, and none can accommodate continuous predictor variables." Isn't that really only true of some approaches for the predefined set? There are groups that use the predefined sets in nested generalized models that can be extended to any complex experimental design and can include confounders and covariates. Maybe not in human genetics but at the very least in *Drosophila* and Arabidopsis. This should probably be tempered.

In the first section on permutation analysis (Table 1). Because of the lack of replication on individuals, VarQTLs in human genetics are a blend of all non-additive sources of variance, especially epistasis and stochasticity. Is there a difference in how epistatic variance or stochastic variance influences the claims made in this section? I ask because mechanistically it would be possible for stochastic variance to be more localized to the gene in question and not spread to other genes while epistatic variance could influence the whole network.

In the section on immunity, what was the reasoning behind limiting their analysis to the 1460 genes that were differentially expressed at the 2 hour time point? I would think that the power of the correlation approach would be that it could get more directly at the interaction term than limit itself to single timepoint selection. This choice should be explained. Especially as I wonder if this choice could create the appearance of decoherence as it does not allow for genes showing a difference at the other time points but not at 24 hours.

The RAP1GAP focus of the final section of the results has me slightly off-kilter. The authors argue for correlation being separable from other traits but then the key example is a gene with a large effect cis polymorphism. This could lead to a change in correlation by shifting genes from regulated by RAP1GAP when RAP1GAP is well expressed to no-correlation when there is low to no RAP1GAP simply because RAP1GAP is not functional. I agree that this is a real observation and does show some utility of their approach. But it would be very helpful to provide some insight of the approaches ability to get at the deeper questions that drive this manuscript. Like some examples of TF-Target correlations that were affected by trans polymorphisms. Or even metabolic gene-metabolic gene correlations that mapped to novel loci. This would help illuminate how much novel information this approach is finding.

I may be wrong but if the authors ideas about decoherence are true, shouldn't this predict that the slope in Figure 3B is less than 1? I.e. there is lower general metabolite to metabolite correlation in the y-axis (metabolic syndrome) than in the x-axis (healthy). As such, it seems that the most direct use of this data to support their conclusion would be to test if the slope is significantly lower than 1 as my interpretation of the model predicts. The rest of the information in this section seems superfluous to the idea it is trying to test.

As a small aside, there has been some work in Arabidopsis on how endogenous single gene mutants can or cannot change the correlational network in response to immune stresses similar to what is proposed in Figure 1A/B and Figure 2. These citations help to support the claim that correlation is a genetically separable trait [1,2]. I'm sure there are likely other similar citations in yeast and *Drosophila*.

1) Mine A, Seyfferth C, Kracher B, Berens ML, Becker D, Tsuda K: The Defense Phytohormone Signaling Network Enables Rapid, High-amplitude Transcriptional Reprogramming During Effector-Triggered Immunity. 2018.

2) Zhang W, Corwin JA, Copeland D, Feusier J, Eshbaugh R, Chen F, Atwell S, Kliebenstein DJ: Plastic transcriptomes stabilize immunity to pathogen diversity: The jasmonic acid and salicylic acid networks within the Arabidopsis/Botrytis pathosystem. Plant Cell 2017, online.

---

## [Author Response]

The reviewers, and I, are convinced of the value of your approach (as one reviewer points out, this is long overdue). We do indeed have a limited understanding of the factors that shape correlations between genes and your work addresses this problem. In particular your observation that decoherent metabolites are good predictors is noteworthy, as is your successful identification of 'correlation-QTLs'. Though perhaps it is not surprising that the degree of correlation is under genetic control, it's certainly good to have that confirmed and analysed, as you have done here. Altogether, your paper represents an important advance.

We thank the reviewers and editor for their enthusiasm and insightful comments.

I want to highlight one issue, to do with unacknowledged confounds. As one of the reviewers points out, the inclusion of expression data from multi-cellular tissues like blood is problematic, and I expected you might incorporate into your method one of the more recent analytical approaches to decompose the contributions (e.g. Nat Methods. 2017 Feb 28;14(3):218-219. doi: 10.1038/nmeth.4190). Would that improve the signal?

We agree that gene expression is cell type specific, and analyses of whole blood transcriptome data can thus be problematic. For our correlation QTL analyses (performed using whole blood expression data) there are two potential concerns, described below.

Concern #1: Whole blood gene expression levels are predicted by cell type composition. This is almost certainly true; however, while the presence of cell type effects alone may add noise and reduce our power to map correlation QTL, they cannot produce false positives unless they are connected to genotype. We demonstrate this through new simulations presented in revised Table 1. Specifically, we performed simulations where a variable that is random with respect to genotype affects the mean or variance of gene expression levels; in practice, this variable could be cell type composition, or an environmental or batch effect (see revised Table 1, paragraph two of subsection “Simulations reveal power to detect sources of variance in correlation across many scenarios”). In addition, to explore the effects of cell type composition in the NESDA dataset, we performed deconvolution and reanalyzed our correlation QTL after regressing out inferred cell type proportions from the gene expression data. Here, we find that our correlation QTL results are robust to correction for cell type heterogeneity (see new Figure 4—figure supplement 5).

Concern #2: False positive correlation QTL are produced by genetic effects on cell type heterogeneity. Specifically, if genotype affects cell type proportions, and the pair of genes tested for correlation QTL are expressed in a cell type specific manner, then individuals of a given genotype would have high expression levels of the two focal genes (and thus high values of their product) because of variation in cell composition, rather than genotypic effects on correlation. For the correlation QTL pipeline we used in the main text, this scenario would produce false positive correlation QTL. To examine this possibility in the NESDA dataset, we used inferred/deconvoluted measures of cell type heterogeneity to confirm that the correlation QTL we identified are not in fact ‘proportion QTL’ (i.e., we confirm that SNPs that affect correlation do not also affect cell type proportion). These results are presented in new Figure 4—figure supplement 6 and Supplementary file 9 (see also subsection “Genetic effects on cell type proportions”). Finally, it is worth pointing out that this source of error is not unique to correlation QTL, but is a common issue in even the most rigorous eQTL studies that use heterogenous tissues: specifically, if a gene is expressed in a cell type-specific manner, then any SNP affecting cell type proportion will be identified as a false eQTL.

We have greatly expanded the Materials and methods section titled ‘Cell type heterogeneity-related confounds’ to address the concerns described here, as well as a question from reviewer #1 about the relationship between cell type heterogeneity and BMI (see below).

Separate reviews (please respond to each point):

Reviewer #1:

[…] My first comment goes to the transparency of the description of CILP and its relationship to canalization. I think the way it is presented is overly complicated. If I am not mistaken, CILP is simply a t-test (or potentially ANOVA) of the Pearson correlation coefficients for all pairs of biomarkers in an analysis. The formula at the start of the Results section is that quantity, but then you say it is correlate with some random variable x. In actuality x is a fixed effect in all your comparisons, and it would be clearer to call it a test of association with x since it is the biomarkers that are being correlated. Furthermore, in Figure 3 you switch to the Spearman correlation, which I can see being appropriate, but is at odds with the rest of the description.

We agree that x is a fixed effect (i.e., a predictor variable of interest) in our case, and have changed the language in the first paragraph of the Results section accordingly. However, there seems to be some confusion as CLIP does not involve a t-test on Pearson correlations, but instead uses the product of two traits to generate a vector of correlation values that are specific to each individual. This vector is then modeled as a function of some predictor variable of interest. We’ve added additional text to the Introduction (third paragraph) to clarify the specifics of the method. We hope this clears up the confusion. We note that we do use Pearson correlations in Figure 3, but only for visualization.

Also to this point, canalization is commonly used in reference to increase in variance, which I can see could lead to de-coherence, but that is neither necessary nor sufficient, and this could be remarked upon.

This is a good point, we’ve added text to the Discussion (paragraph four) regarding this issue.

My second comment has to do with the effect of unaccounted for factors on covariance. All transcriptomes and metabolomes have a high correlation structure, due to a combination of trans-acting influences, environmental influences, and mixtures of cell types. As far as I can see, the opening simulation study does not incorporate overall correlation, it only models pairwise effects due to cis-acting variation and correlation-QTL as defined by the authors. This should be acknowledged: it is extremely hard to model true transcriptional variation, but I suspect it makes a big difference.

We agree that many factors influence whole blood transcriptome data, including environmental variables, batch effects, and cell type composition. The end result of all of these inputs (if they are orthogonal to the predictor variable of interest) would be to affect the mean or variance of each transcript and potentially reduce power. To explore this idea, we simulated a predictor variable (e.g., a batch or environmental effect) that affects the mean and variance of gene expression levels and evaluated our power to detect correlation QTL (see revised Table 1, subsection “Simulations reveal power to detect sources of variance in correlation across many scenarios”). In addition, we explicitly simulated batch effects on transcriptome correlation structure, and assessed our ability to detect variation in correlation (Figure 1—figure supplement 2). These new simulations confirm our intuition: batch or environmental effects that change the mean or variance of gene expression levels, or that affect correlation structure, can reduce power. However, these effects do not produce false positive results if the batch/environmental effect is orthogonal to the predictor variable of interest.

(Parenthetically, why call the cis effects b1 and b2, and the correlation effects b – another symbol for the latter would also be less confusing).

We have changed these symbols.

Regarding the blood gene expression datasets, in the Materials and methods you consider the possibility that cell-type mixtures matter but don't find any influence of available cell counts. However, I can tell you from extensive experience that the major impact on whole blood gene expression in relation to BMI is reticulocyte count, which surprisingly is unrelated to RBC count, but if you fit Axis 2 from Preininger et al. (PMID: 23516379), renamed as Axis R in PMID: 29950172 and PMID: 30223463 and shown to strongly associate with BMI in PMID: 25300922 and PMID: 23162571 and countless other studies we've looked at, it is basically reticulocyte count. I certainly am not asking you to reference these studies (!) but do suggest you fit Axis 2 and see if it has an effect, because it is one of the major components of whole blood expression. I agree that the other Axes that correlate with B and T cell or Neutrophils only weakly associate with BMI.

Thanks for this suggestion. We note that our motivation for identifying BMI-associated genes was simply to limit the search space for correlation QTL mapping. While we agree we could more robustly identify BMI-associated genes if we accounted for cell type proportion, exploring BMI effects on transcription is not our main goal. Nevertheless, to understand how robust our set of BMI-associated genes are, we fit Axis 2 and found that it does indeed correlate with BMI (described in Supplementary file 8; subsection “Cell type proportion associations with BMI”). We’ve also added new analyses where we use deconvolution to estimate the proportion of 22 leukocyte cell types, and we checked whether these more fine-grained measures of cell type heterogeneity influence BMI. In general, the transcripts we originally identified as BMI-associated are still significant after accounting for cell type, though some of our original BMI-transcript associations do appear to be mediated by Axis 2/reticulocyte count variation (see new Figure 4—figure supplement 5 and Supplementary file 8, see also subsection “Cell type proportion associations with BMI”).

Minor Comments:Here are a few more minor comments:Figure 2. Panel (C) described in the Legend has been removed from the version in the manuscript.

We’ve fixed this mistake.

LPS study: Need to reference the Fairfax et al. dataset, in the text (you don't do so until the Discussion). Also note that they describe a number of eQTL that have significantly different effects at BL, 2hr LPS, and 24hr LPS, some switching between the two exposure times. Do your correlation QTL tend to be the same, or overlap with, those instances?

This is an interesting question. We’ve added new analyses describing the overlap – which we find to be minimal – between genes that are differentially correlated as a function of condition (LPS versus baseline) at one time point but not the other and (i) genes that are differentially expressed by condition at one time point but not the other and (ii) genes that have condition-specific eQTL at one time point but not the other (subsection “Immune challenges results in decreased co-regulation of gene expression”). We’ve also added a citation to Fairfax et al. earlier in the text.

Metabolic study. It is not clear from the text whether the results in Figure 3A to 3C are from t0 or from a combined analysis. I suspect just t0, please clarify. But I am also concerned that there may be inflation of the prediction of disease progression in Figure 3D due to use of the discovery dataset in the testing phase (winner's curse), especially if you used t1 and t3 in the discovery. (There will in any case be a bias due to correlations across times within individuals). I'd rather see a more robust analysis that uses a subset of the individuals and then predicts expression in the remainder – there should be sufficient sample size for this.

We agree that something like 10-fold cross validation could be of use here, but even though our overall sample size is large, there are a very small number of individuals in the class we are most interested in (i.e., individuals that are healthy at the first time point and develop metabolic syndrome at the third time point). More importantly, we believe the analyses as they are are robust, as they only used metabolite data from the first time point: there is no leakage of information across time points. Specifically, we identified metabolites that were differentially correlated between healthy and metabolic syndrome individuals at the first time point (using t0 data only), and then asked whether individuals that developed metabolic syndrome later on (at t3) already showed changes at these metabolites at t0. We’ve clarified this key point in the Materials and methods (subsection “Predictive power of metabolite pairs that lose correlation following the onset of disease”) and Results (subsection “Metabolic syndrome affects co-regulation of particular metabolite classes”).

Subsection “Identification of disease-associated variation in metabolite correlation”, instead of stating the percentages of the subset of genes (74.4 and 25.6), state the percentages of all genes, it will still be 3:1. Is there any indication that the transcripts that encode the enzymes that regulate the de-coherent metabolite sets are also de-coherent? I guess you may need data of both types from the same individuals, or from liver.

This is a good point and this would indeed be our prediction. We actually tried to address this question, but unfortunately, to our knowledge, there isn’t a comprehensive database of genes/enzymes that specifically regulate the metabolites we’re using. Thus, we cannot currently answer this question, but agree that it is an important question for future research.

Regarding the correlation QTLs in NESDA, the first thing I think you should report is what fraction of them are also main effect cis-eQTL for one or the other transcript in each pair.

We’ve added this information to the Results (subsection “General characteristics of correlation QTLs”) and have also added a new figure (Figure 4—figure supplement 1) and table (Supplementary file 5) to provide additional information on this question.

Concerning RAP1GAP specifically, is it a trans-eQTL hotspot (I have some vague recollection that it may be, in blood).

We’re not exactly sure what the reviewer is thinking of. The best evidence we’re aware of for trans eQTL involvement is from Fehrmann et al., 2011. They show that a handful of SNPs associated with mean corpuscular volume are also trans eQTL for RAP1GAP expression, though many of these effects are quite weak (up to FDR<0.5). We’ve added this reference in our section on RAP1GAP (subsection “Potential mechanisms generating correlation QTL”).

The enrichment statistics for its binding sites do not convince me at all: P<0.03 is a bit of a joke, and fibroblasts are not relevant, I don't think that result helps at all.

We agree that fibroblasts are not an ideal tissue for understanding biological processes in blood, but we are limited by the available data. Our main goal with this analysis was to provide a potential mechanism for the correlation QTL we uncovered, and we’ve opted to leave the results in place for this reason. However, we have added text to highlight limitations and caveats of this analysis (subsection “Annotation and analyses of correlation QTL”).

Regarding the TF enrichment, I think you need to define what you mean by "involved in" since we now know that the locally regulated gene is usually not the closest one.

The TF-target gene associations we used in our analyses are from a published database (Han et al. 2015), we did not define TF-target gene associations ourselves. TRRUST was created through sentence-based text-mining followed by manual curation. We’ve edited paragraph two of subsection “General characteristics of correlation QTLs” to make it clear that by ‘involved’ we mean there is an entry in TRRUST defining a given gene as being a target of a given TF.

Finally, it would be interesting to know what happens to the expression in individuals who do not fall into the Healthy or MetS classes. Such ambiguous cases may have even more extreme de-coherence as they are transitional, at least at one of the time points. If the result is interesting it could be worth reporting.

This is an interesting suggestion we had not considered. We went back and reran our analyses with 3 classes (healthy, metabolic syndrome, ambiguous) instead of 2 classes, and compared the results with the analyses presented in the main text. In general, the results are similar, though there is one new functional group of metabolites (fatty acids) that is enriched for differential correlations when we include the third class. These new results are described in the final paragraph of subsection “Metabolite data and metabolic syndrome classification”.

Reviewer #2:

[…] On a general note, the paper is written in the general where this approach and its inferences are possibly of use in non-human systems. However, there isn't an obvious effort to discuss work in other systems, Drosophila, yeast, Arabidopsis, Maize, etc, that might influence the ideas and conclusions of this paper. There is similar work in these systems (See below) that touches on these topics and if the goal is really a broad audience they should be cited. If the goal is just a human audience than every term should be caveated by the word human, i.e. evolutionary genetics should be human evolutionary genetics, biology should be human biology, etc. Human Genetics can benefit from reading and acknowledging the work in other systems especially as Human Genetics greatly benefits from the of excellent young senior authors who come from other systems like Drosophila.

Thanks for bringing this issue to our attention, we do mean for our work to appeal and apply to a broad audience. In addition, we appreciate that there is similar and important previous work coming from non-model systems. We have revised the Introduction (paragraph two) and Discussion (first and last paragraphs) to pull in more work from non-human systems, and to discuss the utility of our method for asking questions in non-human systems.

In the Introduction, the authors say "asking whether predefined sets of genes are differentially co-expressed between groups. Almost universally, these approaches are only designed for comparisons between two groups, and none can accommodate continuous predictor variables." Isn't that really only true of some approaches for the predefined set? There are groups that use the predefined sets in nested generalized models that can be extended to any complex experimental design and can include confounders and covariates. Maybe not in human genetics but at the very least in Drosophila and Arabidopsis. This should probably be tempered.

There is indeed a rich literature on the analysis of differential correlation. We previously pointed to a review on this topic, but have now added primary references to place our work in context (Introduction paragraph two), including many references from model organism work. However, to our knowledge, these studies still rely on population-level estimates of correlation coefficients as opposed to the individual-level metric we describe in this manuscript. Without an approach to model an individual-level metric, individual-level covariates (e.g., age, sex) cannot be accounted for. We have revised the Introduction to make it clear that this is the type of covariate/confounder accounting we are referring to.

In the first section on permutation analysis (Table 1). Because of the lack of replication on individuals, VarQTLs in human genetics are a blend of all non-additive sources of variance, especially epistasis and stochasticity. Is there a difference in how epistatic variance or stochastic variance influences the claims made in this section? I ask because mechanistically it would be possible for stochastic variance to be more localized to the gene in question and not spread to other genes while epistatic variance could influence the whole network.

We agree that genetic effects on gene expression variance (i.e., varQTLs) are an important potential mechanism that may contribute to or interact with corQTL. Our simulations (Table 1) show that the presence of varQTLs do not produce false positive corQTL, though they do substantially reduce power. To understand the role of varQTLs in our empirical data, we used dGLM to ask whether any of the corQTL we identified are also varQTL. We find a handful of examples, and describe these results in subsection “General characteristics of correlation QTLs”.

In the section on immunity, what was the reasoning behind limiting their analysis to the 1460 genes that were differentially expressed at the 2 hour time point? I would think that the power of the correlation approach would be that it could get more directly at the interaction term than limit itself to single timepoint selection. This choice should be explained. Especially as I wonder if this choice could create the appearance of decoherence as it does not allow for genes showing a difference at the other time points but not at 24 hours.

We agree that analyzing correlations among all genes, rather than just a subset, would be ideal. However, the sample size for the Fairfax et al. dataset is rather modest (n=214 individuals assayed across time points). Given that we tested for LPS effects on correlation across all possible gene pairs in a given set, interrogating the entire transcriptome would result in too many tests for the sample size at hand (e.g., interrogating pairwise correlations across 10,000 genes would involve 49,995,000 tests). In the future, larger datasets will be very useful for exploring correlations across the entire transcriptome. We point out the degree to which the testing burden scales with the number of genes analyzed in paragraph six of the Discussion.

The RAP1GAP focus of the final section of the results has me slightly off-kilter.

We focused the final section on RAP1GAP specifically, because 425/484 (88%) of the correlation QTL we identify involve this gene. For the subset of genes we tested for correlation QTL, a handful of RAP1GAP cis eQTL appear to be master regulators of large scale correlation structure. We felt it was therefore necessary to provide information about this gene and a potential mechanism. We have however shortened this section substantially, and have added additional examples to highlight other mechanisms that may generate correlation QTL (subsection “Potential mechanisms generating correlation QTL”).

The authors argue for correlation being separable from other traits but then the key example is a gene with a large effect cis polymorphism. This could lead to a change in correlation by shifting genes from regulated by RAP1GAP when RAP1GAP is well expressed to no-correlation when there is low to no RAP1GAP simply because RAP1GAP is not functional. I agree that this is a real observation and does show some utility of their approach. But it would be very helpful to provide some insight of the approaches ability to get at the deeper questions that drive this manuscript. Like some examples of TF-Target correlations that were affected by trans polymorphisms. Or even metabolic gene-metabolic gene correlations that mapped to novel loci. This would help illuminate how much novel information this approach is finding.

We are glad that the main example highlights the utility of our approach. We have added new text (subsection “Potential mechanisms generating correlation QTL”) to highlight additional examples, such as trans regulation of correlation for: (i) gene pairs annotated for involvement in the same GO terms (e.g., GLRX5 and RPS29); (ii) TF-TF target gene pairs (e.g., GATA2 and RAB24); and disease-associated genes pairs (e.g., SLC4A1 and TPM1). We note that in many cases the exact mechanism linking two genes involved in a correlation QTL is unclear, but our approach does map many novel loci regulating key disease and metabolic genes that could be mechanistically investigated in future work.

I may be wrong but if the authors ideas about decoherence are true, shouldn't this predict that the slope in Figure 3B is less than 1? I.e. there is lower general metabolite to metabolite correlation in the y-axis (metabolic syndrome) than in the x-axis (healthy). As such, it seems that the most direct use of this data to support their conclusion would be to test if the slope is significantly lower than 1 as my interpretation of the model predicts. The rest of the information in this section seems superfluous to the idea it is trying to test.

Yes, your intuition is correct, and we did not initially test this. We’ve added text to point out that the slope of the line in Figure 3B is <1 for all points (β=0.814) and especially for just significant points (β=0.597; see revised legend for Figure 3).

As a small aside, there has been some work in Arabidopsis on how endogenous single gene mutants can or cannot change the correlational network in response to immune stresses similar to what is proposed in Figure 1A/B and Figure 2. These citations help to support the claim that correlation is a genetically separable trait [1,2]. I'm sure there are likely other similar citations in yeast and Drosophila.1) Mine A, Seyfferth C, Kracher B, Berens ML, Becker D, Tsuda K: The Defense Phytohormone Signaling Network Enables Rapid, High-amplitude Transcriptional Reprogramming During Effector-Triggered Immunity. 2018.2) Zhang W, Corwin JA, Copeland D, Feusier J, Eshbaugh R, Chen F, Atwell S, Kliebenstein DJ: Plastic transcriptomes stabilize immunity to pathogen diversity: The jasmonic acid and salicylic acid networks within the Arabidopsis/Botrytis pathosystem. Plant Cell 2017, online.

Thanks for pointing us to this work. We have added these references, as well as several new additional references to additional model organism work, in the final paragraph of the Discussion section.